# Conjunctive encoding of exploratory intentions and spatial information in the hippocampus

Yi-Fan Zeng [1,2], Ke-Xin Yang[2], Yilong Cui[1,2], Xiao-Na Zhu[2], Rui Li[1,2], Hanqing Zhang [1], Dong Chuan Wu [3,4], Raymond C. Stevens[1,2], Ji Hu [1,2] & Ning Zhou [1] ✉

The hippocampus creates a cognitive map of the external environment by encoding spatial and self-motion-related information. However, it is unclear whether hippocampal neurons could also incorporate internal cognitive states reflecting an animal's exploratory intention, which is not driven by rewards or unexpected sensory stimuli. In this study, a subgroup of CA1 neurons was found to encode both spatial information and animals' investigatory intentions in male mice. These neurons became active before the initiation of exploration behaviors at specific locations and were nearly silent when the same fields were traversed without exploration. Interestingly, this neuronal activity could not be explained by object features, rewards, or mismatches in environmental cues. Inhibition of the lateral entorhinal cortex decreased the activity of these cells during exploration. Our findings demonstrate that hippocampal neurons may bridge external and internal signals, indicating a potential connection between spatial representation and intentional states in the construction of internal navigation systems.

Every day, as we routinely walk by a street corner statue, our hippocampal place cells fire, letting us know that we are passing by that location. However, what happens when we spontaneously choose to stop and take a good look at the statue? At that moment, is it possible that a group of neurons recognizes our intention and encourages us to pause and contemplate the artwork? These neurons could be aware of where we go and what we want to do.

Neurons in the hippocampus and entorhinal cortex exhibit spatial navigation-dependent activity, supporting the formation of a cognitive map of space[1]. Several cell types involved in cognitive map formation are strongly correlated with navigational parameters, such as spatial position, head direction, or relative distance to landmarks, which correspond to individual's objective movement in their environment[2,3]. One such cell type is the hippocampal place

cell, which fires in response to a specific location in a human or animal's environment as they travel through that location[4].

In rodent experiments, the firing of place cells reliably signals the animal's location and is previously thought to be primarily influenced by positional or feature-in-place information[5]. This means that these cells fire whenever the animal travels to a particular location, or when a feature is added or removed from that location[6]. The feature may be a simple signal, such as a visual or olfactory cue, or a complex stimulus, such as a reward. Later studies have revealed that place cells can be modulated by task contingencies, such as when animals attend to salient cues[7], aim at different intended routes or destinations[8-10], or are motivated by rewards. When a feature, particularly a reward, is introduced into the environment, the animal is attracted to the goal and moves toward it. Under these circumstances, the firing of place cells

[1]iHuman Institute, ShanghaiTech University, Shanghai 201210, China. [2]School of Life Science and Technology, ShanghaiTech University, Shanghai 201210, China. [3]Neuroscience and Brain Disease Center, Graduate Institute of Biomedical Sciences, China Medical University, Taichung City 404333, Taiwan. [4]Translational Medicine Research Center, China Medical University Hospital, Taichung City 404333, Taiwan. ✉e-mail: zhouning@shanghaitech.edu.cn

could be modulated in several ways, including an increase in place field numbers at the goal location[11–13], an increase in firing rates of place cells along the trajectory towards the goal[9,10,14], and the emergence of cells that represent the vectorial distance to the goal[15–17] or cells that fire exclusively at the goal[18]. The close association between reward cues and their spatial representation reflects how place cells are modulated by strong sensory inputs that are essential for an animal's survival.

However, even in the absence of strong sensory cues or reward-seeking behavior, animals may exhibit different behaviors despite following identical moving paths and being exposed to unchanged environmental features. They may either passively move along the path or halt to explore their surroundings, depending on their spontaneous choices and intended actions. Such behaviors are primarily determined by the animal's exploratory intentions and are not influenced by rewards or changes in environmental cues. A potential solution to guide exploratory behaviors at specific locations could involve a specialized group of cells that encode both investigatory intentions and spatial information. However, it is unclear whether such exploration-selective place cells exist.

In this work, we designed an exploration task in which the mice were habituated to an environment with fixed cues. Using miniature microscope imaging, we recorded activity in dorsal CA1 neurons expressing the calcium indicator GCaMP6f. Mice learned to move along a trajectory with a consistent direction and were allowed to freely pause and explore previously encountered objects. We then recorded a population of hippocampal neurons to investigate their representation of spatial information and selectivity for exploration behaviors. Finally, we examined the role of the lateral entorhinal cortex (LEC) in the joint representation of intention and spatial information in these cells.

## Results

### Task design to differentiate between exploration and non-exploration behaviors

To investigate the activity of dorsal CA1 neurons during spontaneous exploration tasks in mice, we trained them to run anticlockwise on the circular track in either a square- or circular-shaped maze (Fig. 1a), as previously studied in rats[19]. Upon completion of each lap, a reward was delivered at a fixed point, and the mouse encountered familiar objects at fixed positions, which they could opt to investigate the object or run without exploration. Notably, the reward was delivered at a different location from the objects and was not associated with the animal's

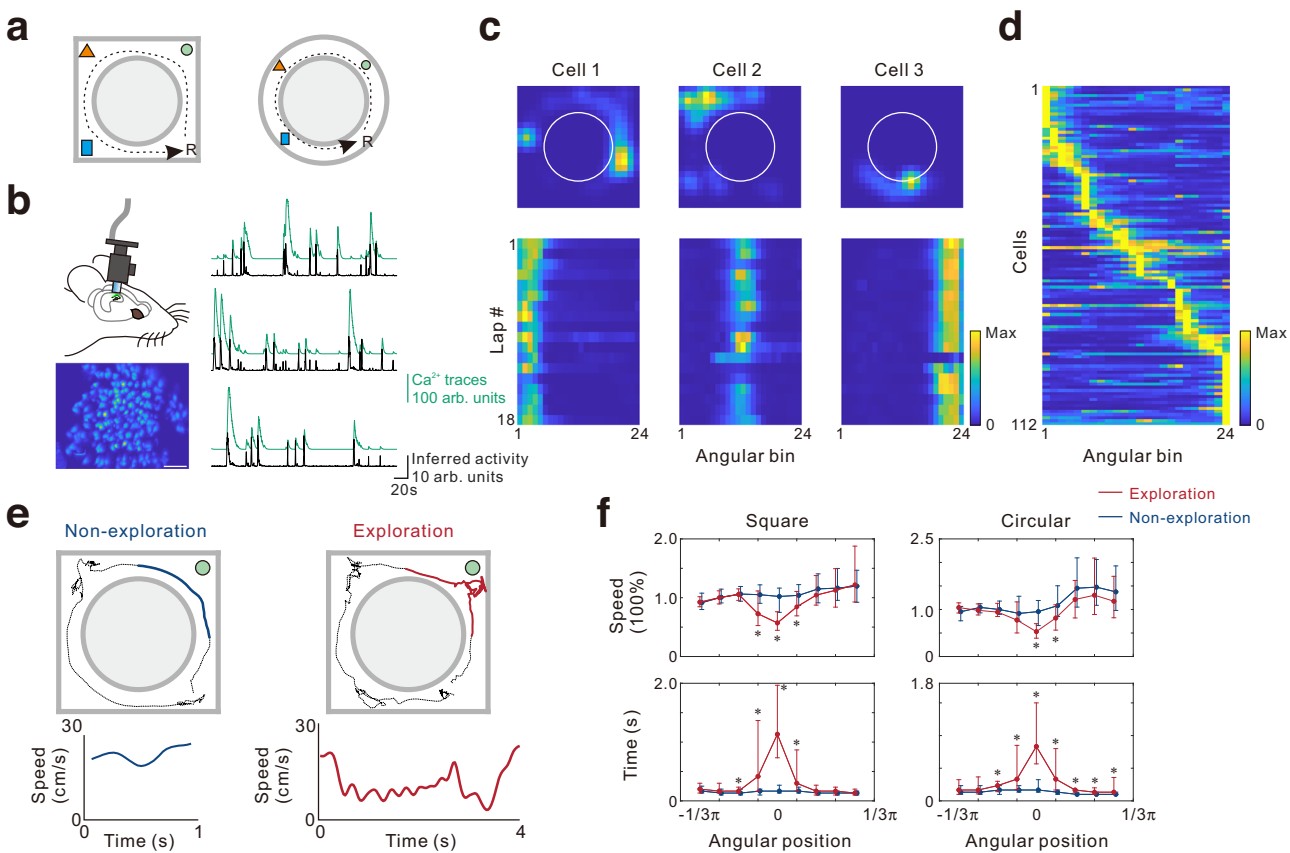

**Fig. 1 | Experimental design for studying intentional exploration behaviors. a** Schematic representation of the experimental apparatus in which mice were trained to run anticlockwise in annular mazes. A reward was delivered at a fixed position (R) and objects were placed at fixed positions, as illustrated. **b** The diagram showing miniature microscope imaging of dorsal CA1 regions of the hippocampus. Bottom left: the spatial footprint of all activated cells during a sample recording session (scale: 50 μm). Right: Ca²⁺ fluorescence traces (green) and deconvolved neuronal activity (black) of three representative cells. **c** Gaussian-smoothed place fields of three example cells that met the criteria of classical place cells. **d** Lap-average activity in a representative experiment, sorted by the place field centers relative to the reward location. **e** Top: Example laps showing the moving trajectory (dashed line) when the animal explored an object or passed an object without

exploration. For brevity and clarity, only the object under analysis is depicted. Comparing the exploration bout (red) to the non-exploration bout (blue), the exploration bout is characterized by decreased moving speed and increased time. **f** Significant difference in moving speed change (top) and investigatory time (bottom) were found in exploration (red) versus nonexploration bouts (blue) (58 and 96 laps, 9 experiments, 9 mice in the square maze; 43 and 79 laps, 7 experiments, 6 animals in the circular maze). Behavioral parameters were averaged in 24 angular bins after the trajectory was converted into polar coordinates with reference to the center of the maze. The position of each object was set to 0. Data are represented by median and interquartile range. *$p < 0.01$ determined by two-sided Mann-Whitney U test. Data and statistical analyses are reported in the Source Data file.

exploration behaviors. During the task, we recorded the calcium activity of CA1 neurons with miniature microscopes (Fig. 1b). We transformed the track position into polar coordinates using the center of the maze as the reference and calculated the sorted neuronal activity in each lap with angular bins (Fig. 1c, d). Our analysis revealed that approximately 50% of neurons exhibited increased activity at specific locations with a significant spatial information index (in total 1012 cells (9 mice) in the square maze and a total 698 cells (6 mice) in the circular maze).

Rodents typically exhibit exploratory behaviors such as ambulation, nose poking/sniffing, visual exploration, and manipulation of objects[20]. Accordingly, a blinded trained observer manually labeled mouse exploratory behaviors in this study. After sorting animal behavior bouts into exploration and nonexplorations bouts (Fig. 1e), we analyzed behavioral variables. The results indicated a significant decrease in velocity, accompanied by a significant increase in the time spent at the object position during exploration bouts (Fig. 1e, f). Additionally, the trajectories in the time-velocity plots also revealed distinct patterns between exploration and non-exploration bouts (Supplementary Fig. 1). These behavioral characteristics were observed in both the square and circular maze. Together, these findings demonstrate a clear distinction between exploratory and non-exploratory behaviors.

## Identification of neurons simultaneously encoding spatial information and exploratory intentions

Compared to traditionally defined place cells that maintained stable place fields in both exploration and nonexploration laps (Fig. 2a), a separated group of CA1 cells exhibited spatial fields only during exploration (Fig. 2b, c), as if they represented specific locations in an exploration-dependent manner. To test this possibility, we compared the binned activity of each neuron between exploration bouts and non-exploration bouts at the same location. A cell was considered an exploration-dependent cell if the difference in its activity between exploration and non-exploration bouts in any near-object bins exceeded the 99th percentile value of the activity differences in the shuffled data. We identified 107 cells (6.3% of all recorded neurons from 15 mice, Fig. 2e, Supplementary Fig. 2) based on this criterion and referred to these cells the object exploration-dependent place cells (oePCs). Of these, 73.8% had only one activity field, while 26.2% had other place fields unrelated to exploration. The oePCs exhibited significantly higher spatial information in the exploration laps than the non-exploration laps (Fig. 2f). Additionally, 97.6% of oePCs exhibited significant spatial information computed from all laps and thus also met the criteria for traditionally defined place cells. In this paper, we refer to the place cell population excluding oePCs as classical place cells (cPC). Compared to cPCs, whose place fields were evenly distributed across the entire track, the place fields of oePCs were maximally distributed with proximity to the locations of objects, as shown in the activity map sorted according to place field centers (Fig. 2d, e) and distribution of the center of mass (COM) (Fig. 2f). Moreover, cPCs and oePCs differed substantially in the Difference Index (DI), which measured the magnitude of difference between exploration and non-exploration bouts (Fig. 2f).

To investigate the emergence of oePCs, we monitored hippocampal neuronal activity during training sessions in a subset of mice (Supplementary Fig. 3). Our findings indicated a gradual increase in the number of cells meeting the criteria for oePC with extended training sessions. For oePCs identified from the fourth day, their activity was tracked back to the initial three days, revealing that most of these cells did not display significantly higher spatial information during exploration compared to non-exploration laps during this early period. Notably, DI values of these cells significantly exceeded zero starting from Day 2, indicating a predisposition for activation during

exploration compared to non-exploration. Collectively, these results suggest that oePCs exhibit a representation of exploratory behaviors from the early experimental sessions, with their spatial specificity evolving gradually over the course of training.

While CA1 principal excitatory neurons exhibit characteristic spatial tuning, some CA1 interneurons have also been shown to display spatial modulation[21–23]. However, these interneurons typically exhibit continuous firing patterns across the entire spatial track and have broader place fields compared to excitatory neurons[21,24]. To determine whether oePCs primarily consist of excitatory neurons, we conducted an examination of the presence and properties of oePCs by using the AAV2/9-CaMKIIα-GCamP6f, which predominantly infects excitatory neurons (Supplementary Fig. 4a). Our data showed that 84.4% of neurons labeled with CaMKIIα-GCamP6f met the criteria of place cells, with 10.1% of total cell population identified as oePCs. Notably, the percentages of both place cells and oePCs among CaMKIIα-GCamP6f labeled neurons were higher than those observed in hSyn-GCamP6f labeled mice. Importantly, the spatial characteristics, center of mass, and difference index of oePCs remained consistent between CaMKIIα-GCamP6f and hSyn-GCamP6f labeled mice (Supplementary Fig. 4b–e). These findings collectively support the inference that the majority of oePCs primarily consist of excitatory neurons.

## The oePCs are activated before exploration behaviors

As all objects were placed outside of the circular track in the square maze, the animal's trajectory might deviate slightly from the center when they encountered the objects, resulting in an increase in radial distance and a shift in head orientation. An illustrative example is presented in Fig. 3a, b, where an exploration bout exhibited great overlap with non-exploration bouts (Fig. 3a). In a different exploration bout within the same recording session, the mouse moved toward the object, generating off-track positions (Fig. 3b top). Concurrently, the head directions might display varying orientations compared to those observed during non-exploration bouts (Supplementary Fig. 5a, b). Consequently, it was possible that the enhanced activity of oePCs might be attributed to responses to different place fields or head directions in these off-track positions. To rule out these possibilities, we excluded off-track positions during exploration bouts by determining the animal's radial distance (Rho) relative to the center of the maze, which exceeded 2 standard deviations of the Rho values in non-exploration bouts (Fig. 3a, b bottom, see Methods). After the exclusion of off-track positions, in-track activity of oePCs remained significantly higher in the exploration bouts for 94.7% of oePCs in the square maze and 96.9% in the circular maze, while only five cells in the square and two cells in the circular maze were modulated by the off-track positions. Additionally, to address the influence of shifted head directions, we excluded points where the head direction exceeded 2 standard deviations of the mean calculated from the non-exploration bouts at each angular position (Supplementary Fig. 5c). Following this adjustment, oePC activity remained significantly higher in exploration bouts for 97.3% of oePCs in the square maze and 93.8% in the circular maze.

Next, we employed decoding approaches to investigate the temporal relationship between oePC activity and object exploration behaviors. The accurate decoding of exploration versus non-exploration behaviors from oePC activity was observed, showing above-chance significance at angular positions (Fig. 3c left). To mitigate potential influences from the off-track positions, behaviors were decoded from oePC activity after excluding off-track points. Remarkably, precise prediction was maintained, particularly preceding the object location. This analysis indicates that oePCs carry sufficient information to encode the animal's investigatory behavior, and their activity is independent of the off-track positions. In addition, the

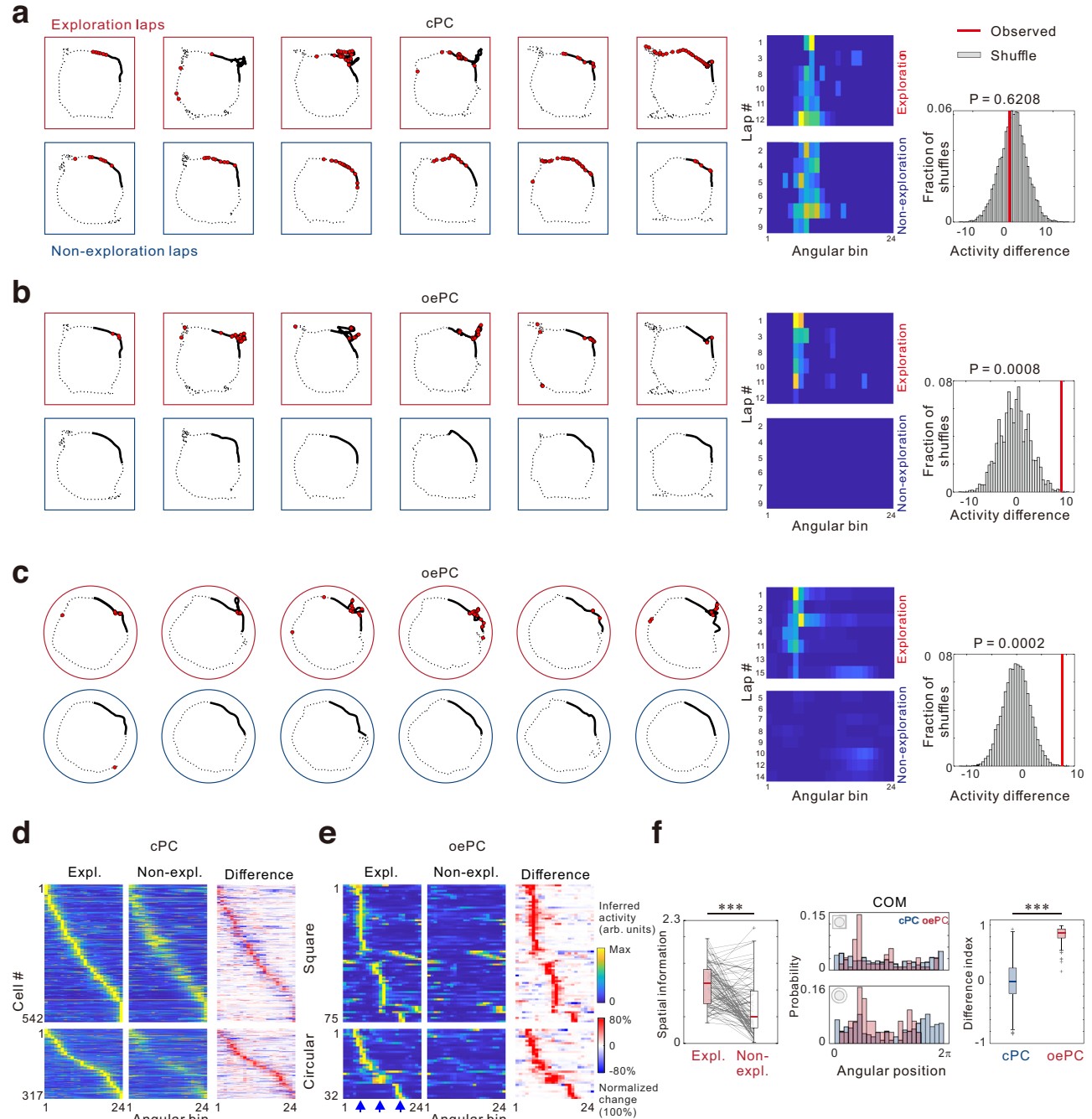

**Fig. 2 | Identification of object exploration-dependent place cells (oePCs). a** A sample classical (cPC) with mouse trajectory and deconvolved spike activity (red). Middle panels show binned activity in exploration (top, red) and nonexploration (bottom, blue) bouts. The right panel shows the observed difference in mean spike rates between exploration (Expl.) and nonexploration (Nonexpl.) bouts and shuffled data using bootstrap methods. **b, c** Same as in **a** but for sample oePCs in the square-shaped (*n* = 75 cells in 9 mice) and circular-shaped maze (*n* = 32 cells in 6 mice). **d** Lap-average activity of cPCs in exploration and non-exploration laps across all mice, sorted by place field centers. Neuronal activity was normalized to the maximum activity of each neuron and sorted according to angular position. The right panel shows difference between the activity map in exploration and non-exploration laps. **e** Same as in **d** but for oePCs. Arrowheads indicate bins closest to the objects. **f** Left, difference in spatial information between exploration (red) and non-exploration laps (white) in 107 oePCs (*P* = 2.41 × 10⁻⁹; two-sided paired Wilcoxon signed-rank test). Middle distribution of center of mass (COM) in the square (top) and circular maze (bottom). Right, Difference Index (DI) in cPCs (blue, n = 859) and oePCs (red, *n* = 107), respectively (*P* = 1.29 × 10⁻⁵⁹; Two-sided Mann-Whitney U test). Box plots show the median (horizontal line), 25–75% range (box) and outliers (whiskers). Data and statistical analyses are reported in the Source Data file.

robustness of these findings was further validated by decoding behaviors with Rho from various angular positions as a control. This analysis revealed the above-chance prediction accuracy at angular bins nearest to the object location (Fig. 3c right), suggesting differences in positions between exploration and non-exploration bouts. However, this difference diminished when off-track positions were excluded, suggesting that the predicted behaviors prior to reaching the object originate from differences in oePC activity rather than the animal's trajectory.

Furthermore, single-cell activity maps for all oePCs during each exploration bout were computed and compared to those during non-exploration bouts, as illustrated for two simultaneously recorded

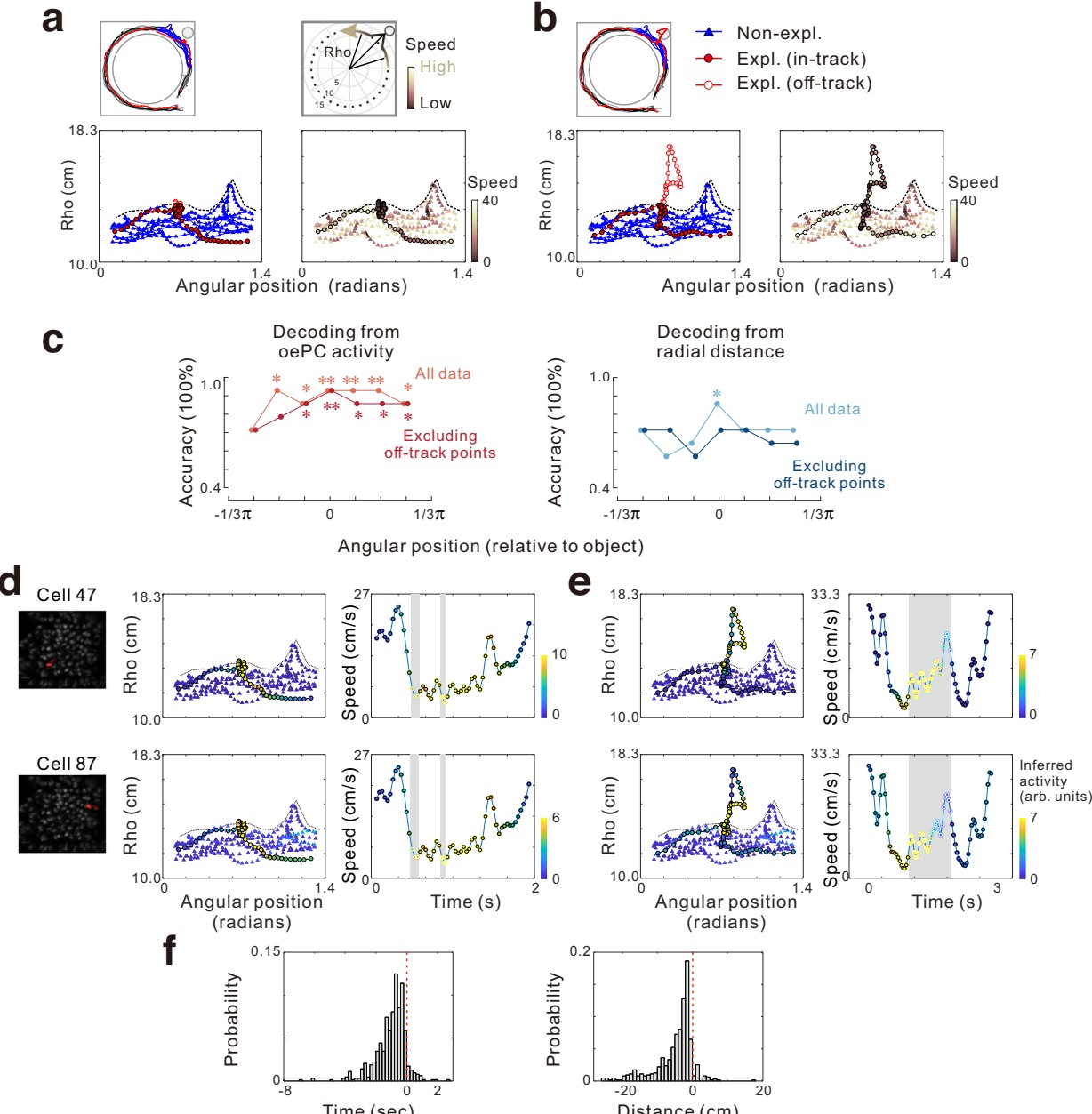

**Fig. 3 | Activity maps of object exploration-dependent place cells (oePCs).**
**a** Sample paths taken by a mouse during a sample exploration bout (Expl., red), which is highly overlapped with the paths during non-exploration bouts (Non-expl., blue). Top, observed paths (left) taken in the maze and diagram (right) showing changes in radial positions (Rho) and speed. Bottom, behaviors (left) and speed (right) along the running trajectory of the mouse plotted on angular-radial coordinates for each frame. The dashed line indicates the boundary between in-track and off-track positions (> mean + 2 × SD). **b** Same as in **a** from the same experiment but for a sample exploration bout in which the mouse ran through off-track positions (red open circles). **c** Decoding accuracies for exploration/non-exploration

behaviors decoded from oePC activity (left) or radial positions (right) at different angular positions relative to the object before (light line) or after excluding off-track points (dark line). *$P < 0.05$ and **$P < 0.01$, decoding significance relative to the shuffled distribution (one-sided, see Methods). **d** Inferred activity of two sample oePCs in the angular-radial position plot (middle) and time-speed plot (right). Left panels show maps of the two cells. **e** Same as in **d** from the same sample cells but for the exploration bout with off-track positions (shaded regions). **f** The time and distance relative to reaching the object by the mouse when oePCs were activated (633 trials × cells). Data and statistical analyses are reported in the Source Data file.

oePCs (Fig. 3d, e). The activity map along the moving trajectory showed that oePCs increased activity before the animal entered the off-track regions. Quantified results revealed that most oePCs were activated before the mouse reached the object with a median time difference of 0.78 s, (25th–75th percentiles 0.40–1.48 s) and a median distance 3.04 cm (1.40–6.64 cm, Fig. 3f). Together, these results indicated that oePCs were activated before the mice arrived at the object location, and the enhanced activity during exploration was not due to the entrance of different place fields.

Several possible explanations for why oePCs activity preceded exploration behaviors can be considered: firstly, oePCs encoded the intention to explore; secondly, oePCs might simply reflect the presence of an object; and thirdly, oePC activity might encode memories associated with the object. To exclude the second possibility, we designed a set of experiments in which the object was positioned outside the track and obstructed from view by opaque partitions (Supplementary Fig. 6). After extensive training and learning the object's location in this setting, mice were allowed to freely explore the

concealed object by passing through a designated door or bypass the door location without exploration (Supplementary Fig. 6a). Notably, the object was visually inaccessible unless the mice chose to pass through the door. Data corresponding to off-track points, where mice exited the track, were excluded (see Supplementary Fig. 6b). Applying the same oePC identification criteria revealed a subset of neurons that increased activity before object exploration. Activity maps of these cells clearly demonstrate activity preceding the mice reached the designated door and visualized the presence of the object (Supplementary Fig. 6c). These oePCs exhibited properties consistent with those identified in the initial experiments (Supplementary Fig. 6d–f). These results collectively provide evidence that oePCs encode the intention of object exploration rather than merely responding to the visual perception of the object.

The third possibility, that hippocampal neurons might encode object-associated memories, has been demonstrated by a previous study wherein rodents, as they explore their environment through head-scanning behaviors, previously silent hippocampal cells could develop new place fields that potentiated through the rest of the session[25]. Following the same procedure to quantitatively assess whether the observed enhancement of oePC activity during exploration laps was also the case, we analyzed cell activity during 85 exploration events in 107 oePCs[25] (Supplementary Fig. 7). However, none of these events met the criteria for a place-field potentiation event. Taken together, our results indicate that oePCs primarily signal the intent to explore, not directly encoding object information or memory.

### The activity of oePCs is unrelated to reward

The animals exhibited a reduced moving speed when approaching the object during exploration, as shown in Fig. 3d, e. This observation led to the possibility that the enhanced activity of oePCs might be solely related to the decrease in speed. To address this concern, we computed the average neuronal activity during various behavioral periods that involved a speed-drop, such as exploration at the place field, slowing down before the reward location, and speed-drop at other locations outside of place fields or reward sites (other events) (Fig. 4a). We found that the activity of oePCs during the speed-drop events related to reward or other events was similar to the mean baseline activity, whereas it was significantly enhanced during exploration-related speed-drop events (Fig. 4b). This indicated that the activation of oePC was not due to the modulation of speed. Next, since the activity of oePCs tended to increased when the mouse slowed down before reaching the object, we conducted quantitative analyses to examine the correlation between speed and oePC activity (Fig. 4c, d). The relationship between speed and oePC activity during exploration exhibited a stronger negative correlation coefficient than during reward or other events. The oePCs showed significantly lower correlation coefficients for exploration periods ($P = 1.76 \times 10^{-13}$ determined using Kruskal-Wallis H Test followed by a multiple comparison test, Fig. 4e). Specifically, 67.4% oePCs exhibited a significant negative correlation coefficient during exploration (percentile correlation of <5% of chance), whereas the percentages of oePCs meeting the same criteria during reward and other events were 32.6% and 38.0%, respectively (Fig. 4e). These findings suggest that the activity of oePC increases specifically during the slowing down before exploration behaviors.

As rodents' exploration behaviors can be triggered by external sensory stimulation, such as novel objects or rewards, exploration may be associated with another cell type known to be modulated by the animal's intrinsic states, the reward-associated (RA) cells. These cells are active at locations prior to or subsequent to reward delivery[18]. To determine whether oePCs could be distinguished from these cells, we identified RA cells as cells with a center of placed fields close to the reward site. We found that 8.5% of cells (out of 1710 cells, 15 mice) were active near the reward site (Fig. 4a). We excluded four RA that

were also identified as oePCs in the following analysis. Although RA cells exhibited significant higher activity during the speed-drop events near the reward site, their activity was similar to baseline activity during intended exploration (Fig. 4b bottom). These cells exhibited stronger negative correlation coefficients during reward-related periods than other periods ($P = 1.39 \times 10^{-6}$, determined using Kruskal-Wallis H Test followed by a multiple comparison test, Fig. 4f). 62.8% of RA cells exhibited activity that was negatively correlated with speed prior to reward, while only 25.5% of the same cell population showed above-chance correlation before object exploration (Fig. 4f).

We further investigated whether oePCs remained active when an object was substituted with a food reward at the same location within a recording session (Fig. 4g). The experiment was initiated by allowing the mice to freely explore the familiar object, enabling the identification of oePCs from both the object exploration and non-exploration laps. In a subsequent lap, we removed the existing object and introduced a food pallet at the exact same location before the mouse reached it (referred to as the reward lap). In the following laps, a reward was released only when the mouse exhibited pausing and exploration behaviors at the same location, with approximated a 50% chance determined at random (referred to as the reward-expecting laps). In the reward-expecting laps, the mice exhibited evident anticipation behaviors characterized by significant deceleration prior to reaching the site and increased time spent in that location (Fig. 4h). However, the previously identified oePCs displayed significantly reduced activity during the reward-expecting laps (Fig. 4i, j). Taken together, these findings reveal that oePCs exhibit low activity in response food rewards, indicating a clear differentiation between these cells and reward-associated cells as distinct cell types.

### The oePCs are not feature-in-place cells

We compared properties of oePCs with previously reported mechanisms by which local objects or object-related exploration behaviors could influence hippocampal neuronal activity. First, landmark-vector cells are more active when the animal approaches objects across different locations and may encode the position of the object as a vector relationship to local landmarks[2,26]. These cells tend to develop multiple place fields in response to objects at different locations or altered fields in response to the displaced object[2]. We have found that all recorded oePCs had only a single place field that was associated with object exploration. To further investigate responses of oePCs to object displacement, we moved the object 0.5, 1, 2, or 4 cm away from the original position (Fig. 5a). Displacement of the object caused significant changes in maximum oePC activity compared to the control, where the activity difference was computed between odd and even exploration laps with a fixed object position (Fig. 5b). The degree of place field alterations, quantified by spatial correlations and shifts in the center of mass (ΔCOM), was in parallel with the distance of object displacement. Accordingly, the number of place fields decreased as the object was moved to a greater distance, with all recorded oePCs showing a dissipation of place fields when the object was moved 4 cm away. The DI decreased significantly when the object was moved by more than 1 cm. These data indicate that oePCs exhibit high spatial specificity and are different from landmark-vector cells. Second, hippocampal neurons may represent conjunctive object-location memory and fire differentially during the sampling of different items in one place[27]. We examined this possibility by replacing the familiar objects with a subset of novel objects while their locations remained the same (Fig. 5c). The spatial stability and exploration selectivity between the conditions with familiar and novel objects were similar to the control (Fig. 5d), indicating that oePCs were not sensitive to object identity or novelty. Similar outcomes were observed when using a subset of more complex objects in this experiment (Supplementary Fig. 8a–c). Interestingly, object replacement induced significant changes in cPC place fields near the object (Supplementary Fig. 8d, e), suggesting that object information might be

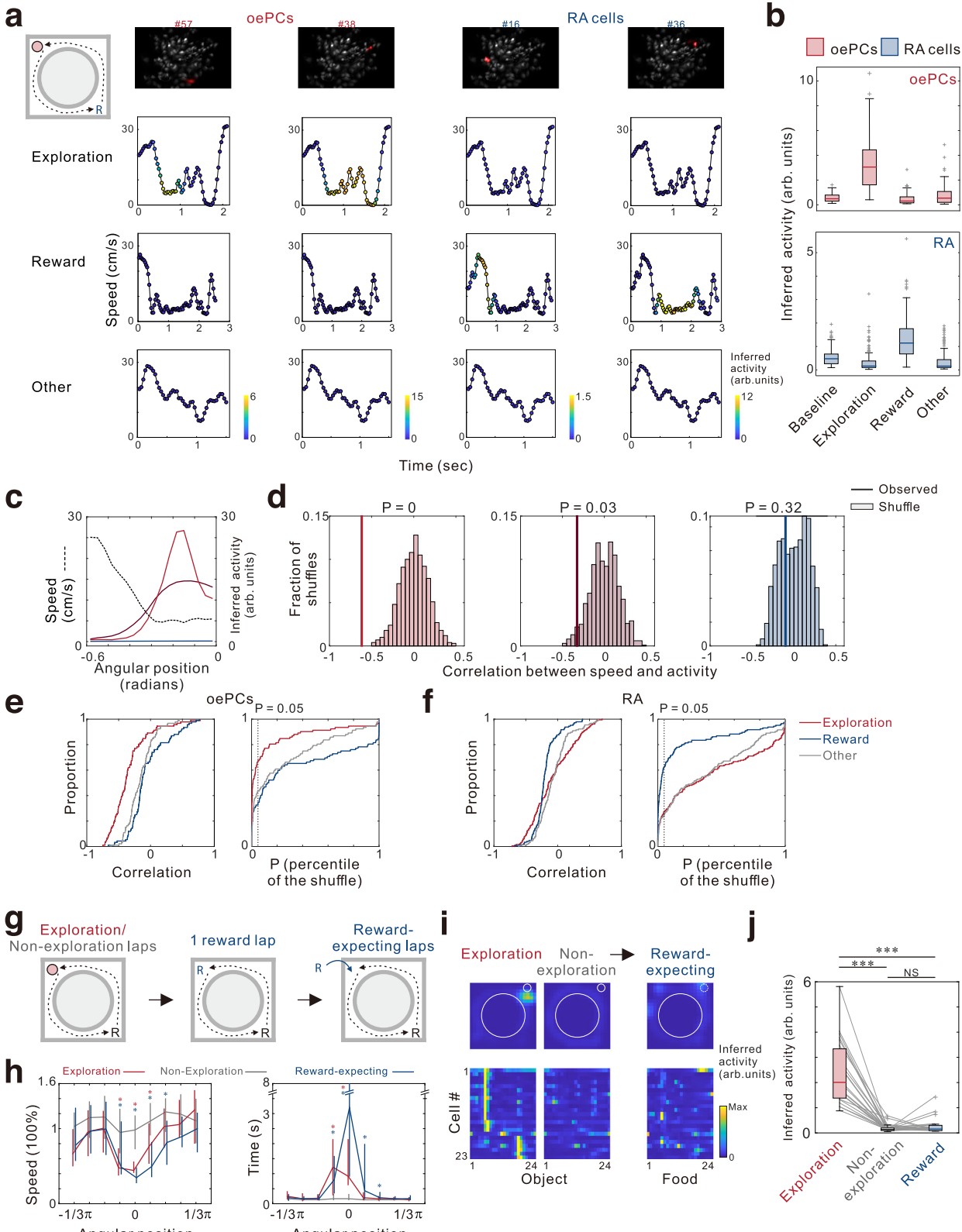

encoded by cPCs rather than oePCs. Third, misplace cells were reported to increase firing after the removal of objects[4,12], likely reflecting the animal's awareness of the alteration of an environmental cue in the place. We found that most oePCs did not activate after the removal of the object (Supplementary Fig. 9), together with the data of the object replacement, indicating that oePCs did not encode information about alterations of feature-in-place signals[6].

As the typical hippocampal place cells are widely recognized as a critical neural substrate for the storage of episodic memories, they can distinguish between environments with different geometrical features through differential coding, and these representations can be maintained over timescales of at least several days[28]. To assess whether oePCs have comparable properties to cPCs, we examined their respective characteristics. First, to evaluate the day-to-day activity of

**Fig. 4 | Increased activity of object exploration-dependent place cells (oePCs) during mouse slowing periods associated with exploration but not reward. a** Inferred activity of sample oePCs (#57 and #38) and reward-associated (RA) cells (#16 and 36) is depicted in the time-speed plot during three representative behavioral bouts from a single experiment. Top: SFP map. **b** Top, mean activity of all oePCs during different periods (n = 92 cells, 15 mice). The activity of oePCs during exploration is significantly higher than the other conditions ($P = 4.36 \times 10^{-37}$; Kruskal-Wallis H Test). Same as in top but for RA cells (n = 145 cells, 15 mice). RA cell activity during reward periods is higher than other conditions ($P = 2.16 \times 10^{-47}$; Kruskal-Wallis H Test). **c** An example exploration-related slowing period showing the negative correlation between speed (black) with the activity of two oePCs (red and dark red) or a RA cell (blue). **d** Correlation coefficients for the cells depicted in (**c**) in corresponding colors. **e,f** Cumulative distributions of correlation coefficients (left) and significance (percentile of the shuffle, right) for oePCs (**e**) and RA cells (**f**)

during exploration (red), reward (blue) and other events (gray). Cells with P < 0.05 are considered significant (dashed line). **g** Experimental diagram illustrating an object replaced by rewards within a recording session. **h** Significant difference in speed change (left) and investigatory time (right) were found in exploration (red) or reward-expecting (blue) bouts compared to non-exploration (gray) bouts (30, 77 and 37 laps, respectively; 6 experiments; 5 mice). Data are represented by median and interquartile range. *P < 0.001 determined by Kruskal-Wallis H Tests followed by multiple comparison tests. **i** Gaussian-smoothed activity maps of a representative oePC (top) and lap-average activity of pooled oePCs (bottom). **j** Higher oePC activity during exploration periods than other conditions ($P = 1.94 \times 10^{-10}$; Kruskal-Wallis H Test, n = 23 cells, 5 mice). Box plots show the median (horizontal line), 25–75% range (box) and outliers (whiskers). *** for P < 0.001, ** for P < 0.01, * for P < 0.05, and NS for no significance. Data and statistical analyses are reported in the Source Data file.

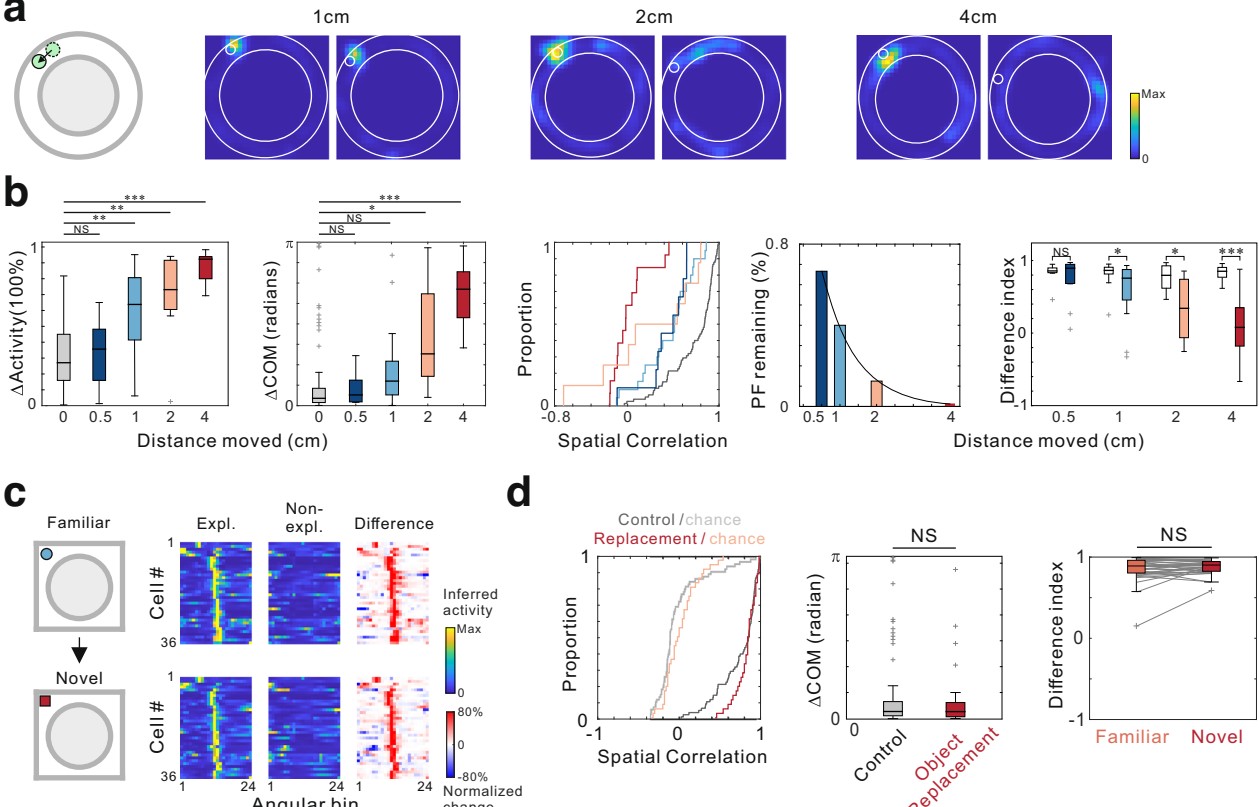

**Fig. 5 | Encoding of location-specific information but not object identity by object exploration-dependent place cells (oePCs). a** Gaussian-smoothed activity maps of three example oePCs before and after the object was moved with different distances. **b** Changes in oePC properties after the object was moved with different distances, including the control (n = 75, gray), 0.5 (n = 9, dark blue), 1 (n = 20, light blue), 2 (n = 8, orange), and 4 cm (n = 13, red) group. Statistics of oePC properties for each group, including activity changes (absolute normalized values, Δactivity), changes in the center of mass (ΔCOM), spatial correlations (determined by Kruskal-Wallis H Test followed by a multiple comparison test), remaining place fields (PF), and difference index (DI) before versus after object movement (two-sided Paired Wilcoxon signed rank test, n = 6 mice). **c** Lap-average activity of oePCs during exploration (Expl.) and non-exploration (Non-expl.) laps and activity difference

during the mice's visits to the same box with the familiar object replaced with a novel object at the same location. Neuronal activity was normalized to the maximum activity of each neuron and sorted by place field centers according to angular position across mice. **d** Spatial property changes in oePCs during object replacement (red) compared to the control condition (gray), including spatial correlation (P = 0.206, two-sided, two-sample Kolmogorov-Smirnov test); ΔCOM (P = 0.717, Two-sided Mann-Whitney U Test), and the difference index (P = 0.293, two-sided Wilcoxon signed rank test; n = 36 in 6 mice). *** for P < 0.001, ** for P < 0.01, * for P < 0.05, and NS for no significance. Box plots show the median (horizontal line), 25–75% range (box) and outliers (whiskers). Data and statistical analyses are reported in the Source Data file.

these cells, we imaged the same field of view when the mouse visited the same maze on two different days (one- or two-day apart) (Fig. 6a). Approximately 67% of oePCs remained active in both paired recording sessions; however, no significant difference was observed in the properties between the temporarily and consistently active oePCs (Supplementary Fig. 10). For all consistently active oePCs, the spatial correlation

and ΔCOM values were comparable to those of cPCs (Fig. 6b), suggesting that the two cell groups exhibited similar day-to-day dynamism in spatial representations. Although the oePCs showed a decreasing trend in DI values, they remained significantly higher than those of the cPCs, suggesting that oePCs were capable of maintaining their ability to differentially encode investigatory intentions across multiple days.

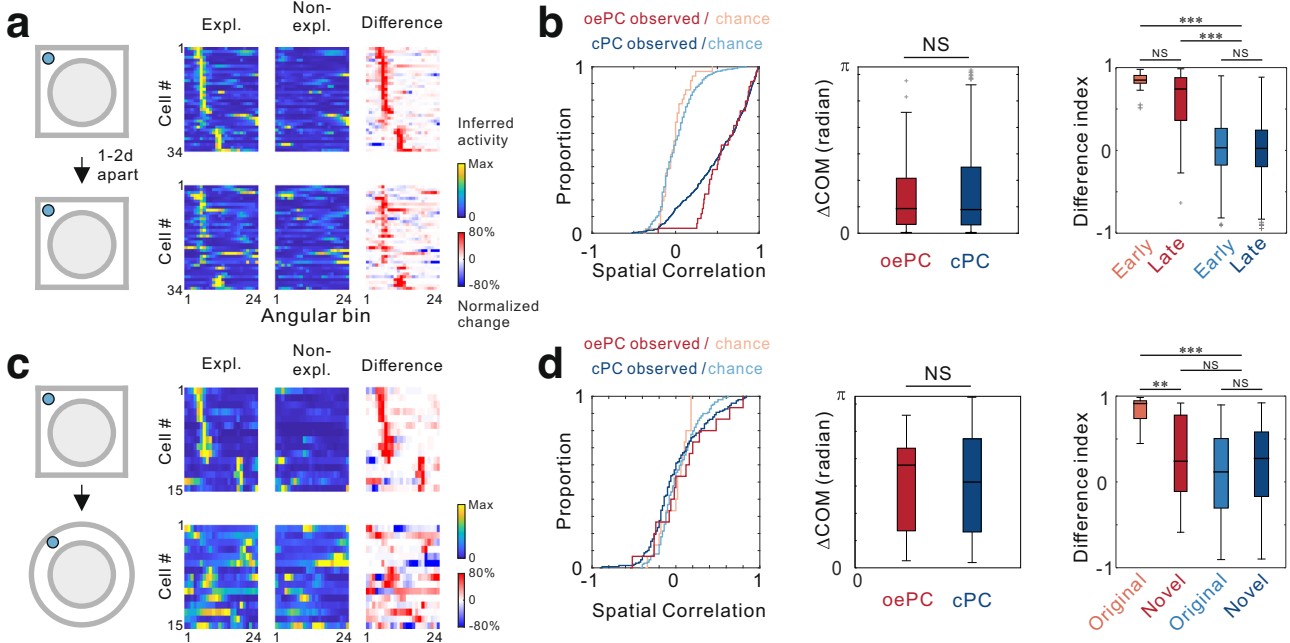

**Fig. 6 | Both object exploration-dependent place cells (oePCs) and classical place cells (cPCs) employ similar dynamic coding mechanisms. a** Lap-average activity of oePCs in exploration (Expl.) and non-exploration (Non-expl.) laps during different days. **b** Statistics for cross-day activity in oePCs (*n* = 34, red) and cPCs (*n* = 393, 8 mice, blue), including spatial correlation (*P* = 0.0497; two-sided, two-sample Kolmogorov-Smirnov test), change of center of mass (ΔCOM; *P* = 0.773; two-sided Mann-Whitney U test), and difference index in the first and second session (P = 2.80 × 10⁻²⁹; Kruskal-Wallis H Test). **c** Lap-average activity of oePCs during visits to different mazes with the presence of same objects. **d** Statistics for oePC (*n* = 15, 5 mice) and cPC (*n* = 120, 5 mice) properties in different mazes, including spatial correlation (*P* = 0.327; two-sided, two-sample Kolmogorov-Smirnov test), ΔCOM (*P* = 0.969; two-sided Mann-Whitney U test), and DI (1.04 × 10⁻⁷; Kruskal-Wallis H Test). *** for *P* < 0.001, ** for *P* < 0.01, * for *P* < 0.05, and NS for no significance. Box plots show the median (horizontal line), 25–75% range (box) and outliers (whiskers). Data and statistical analyses are reported in the Source Data file.

Next, to investigate whether oePCs were capable of remapping in distinct contexts, we placed mice into a novel maze that contained objects from the original maze (Fig. 6c). Approximately 58% of total cells remained active in a different maze, with 71% of oePCs remaining active in both contexts. Among these, 11 out 15 oePCs still exhibited spatial modulation, while 4 cells were not modulated by spatial or exploratory information in the novel context. In these experiments, substantial remapping occurred in both oePCs and cPCs, as the spatial correlations were not significantly different from chance levels in either cell group. Replacing the maze induced a significant reduction of DI in oePCs, suggesting that the exploration-dependent selectivity of oePCs was abolished (Fig. 6d). Together, these findings suggest that similar to cPCs, oePCs utilize a dynamic coding mechanism to represent context-specific spatial information.

### LEC inputs are required for the representation of oePCs

To investigate how the spatial information and behavioral signals converged in the oePC, we aimed to inhibit the LEC, which is a critical input to the hippocampus for encoding non-spatial sensory information[29]. We used the pharmacologically selective designer Gi-protein-coupled muscarinic receptors hM4Di to inhibit bilateral LEC, while mCherry alone was used as a control (Fig. 7a). The efficacy of chemogenetic inhibition of LEC neurons was assessed in brain slices through patch-clamp recordings (Supplementary Fig. 11). In a pair of sessions, we treated both hM4Di-expressing and mCherry-expressing mice with clozapine-N-oxide (CNO) 30 min before the second session and observed no significant changes in their locomotion or exploration behaviors (Supplementary Fig. 12). Next, we examined the effects of inactivating the LEC on the spatial field stability and exploration-dependency of oePCs by computing the relative change in neural activity, spatial correlation, ΔCOM, and DI for each oePC between pairs

of sessions. The CNO treatments did not induce significant changes in these properties in mCherry mice, serving as a control (Fig. 7b, c). However, in hM4Di mice, we observed a significant reduction in neuronal activity during exploration bouts following CNO treatment, but not saline. CNO treatments also significantly impaired the spatial correlation, ΔCOM and DI of oePCs compared to saline (Fig. 7d–f). Additionally, the inhibition of LEC in the same group of animals did not significantly impact the activity of RA cells (Supplementary Fig. 12c, d). These findings indicate that LEC inputs to the hippocampus play a crucial role in the neural representation of oePCs, which are distinct from the responses of RA cells to food rewards.

### Discussion

Our study identified a functional group of hippocampal neurons, which we termed oePCs, that exhibited place fields specific to exploration behaviors. The activity of oePCs differed significantly from previously reported neural representations in hippocampal neurons, which primarily encoded objective information related to the external environment or the animal's self-motion. Notably, the unique encoding properties of oePCs cannot be attributed to spatial information (such as in-track or off-track positions), object information (such as object location, identity or novelty), head direction shifts, or other behavioral variables (such as speed change or reward-seeking). These findings suggest that oePCs represent a unique type of neural coding that may play a critical role in mediating spatial cognitive processing during exploratory behaviors.

Top-down cognitive processes, such as attention and motivation, have been found to modulate the population properties of hippo-campal neurons. Research has shown that manipulating animal attention through changes in environmental cues or salience can impact place field properties, leading to higher stability in place field

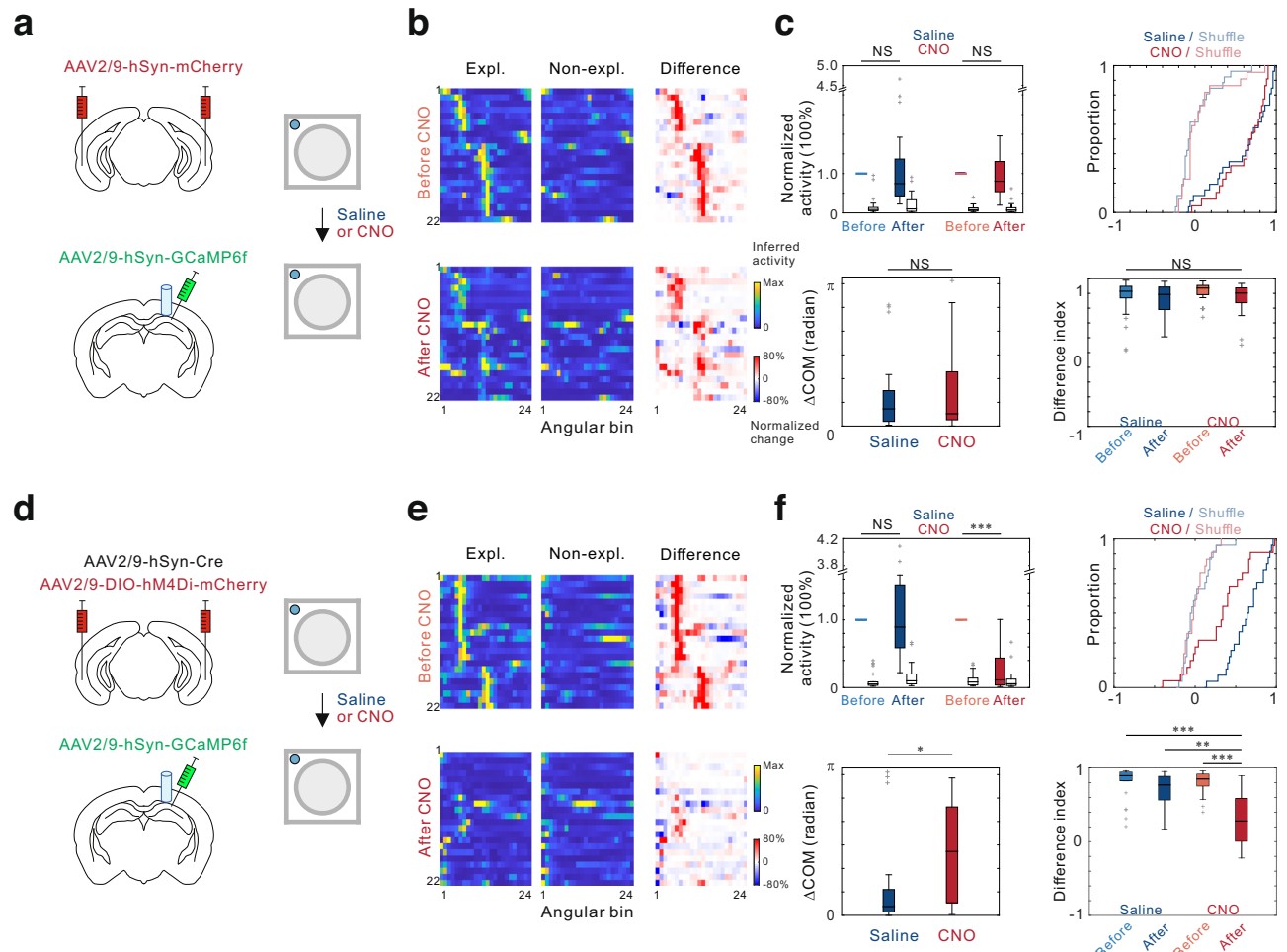

**Fig. 7 | Neural circuit mechanisms underlying object exploration-dependent place cell (oePC) representations. a** Schematic of AAV-hSyn-mCherry injection into the lateral entorhinal cortex (LEC). **b** Lap-average activity map differences between exploration (Expl.) and non-exploration (Non-expl.) laps of oePCs before and after injection of clozapine-N-oxide (CNO). **c** The oePCs show no significant changes in normalized activity during exploration after saline (blue, $P = 0.439$) or CNO treatments (red, $P = 0.390$; two-sided paired Wilcoxon signed-rank test). The spatial correlation ($P = 0.303$; two-sided, two-sample Kolmogorov-Smirnov test), changes in the center of mass ($\Delta$COM; $P = 0.926$; two-sided Mann-Whitney U Test), or difference index ($P = 0.312$; Kruskal-Wallis H test) was comparable after saline ($n = 26$) and CNO injection ($n = 22$ in 6 mice). **d** Similar experiments were performed with bilateral LEC infusion of hM4Di-containing AAV. **e** Lap-average activity map differences between exploration and non-exploration laps of oePCs before and after injection of CNO in hM4Di-expressing mice. **f** The oePC activity exhibited significant decrease after CNO ($P = 4.61 \times 10^{-5}$) but not saline injection ($P = 0.568$; two-sided paired Wilcoxon signed rank test). Significant changes were observed in spatial correlation ($P = 0.0072$; two-sided, two-sample Kolmogorov-Smirnov test), $\Delta$COM ($P = 0.013$, two-sided Mann-Whitney U Test), and difference index after CNO injection ($P = 2.01 \times 10^{-6}$; Kruskal-Wallis H test; $n = 22$ in 5 mice) compared to saline ($n = 24$). *** for $P < 0.001$, ** for $P < 0.01$, * for $P < 0.05$, and NS for no significance. Box plots show the median (horizontal line), 25–75% range (box) and outliers (whiskers). Data and statistical analyses are reported in the Source Data file.

representation, reduced variance in place cell spiking, and increased neural synchronization in the hippocampus[7,30,31]. Changes to feature-in-place signals, such as the removal or replacement of an object, can enhance the firing rate of place cells, as observed in the case of mis-place cells[4,12]. However, these mechanisms typically involve changes to external environmental features, such as adding or removing a feature from a location[6]. Therefore, the observed behavioral modulations of place cell responses in these studies may not only reflect changes in attentional states but also an updated code that integrates alterations in external environmental features, making it difficult to determine whether behaviors initiated by internal top-down signals can also be represented in single hippocampal neurons. In this study, we examined hippocampal neuronal activity at the same location during exploration and non-exploration bouts within a well-habituated environment, with the only different variable being the animal's intention – whether actively exploring or passively traversing the field. The oePCs were found to exhibit a spatial representation that is dependent on the animal's exploratory intention, firing not only before

reaching the target location but even when the object was not directly visible. Therefore, oePCs conjunctively encode both external and internal variables in spatial navigation. These cells might be a potential neural substrate through which rodents intentionally control their hippocampal activity and volitionally access the cognitive maps[32].

Engaging animals in different task contingencies, whether they involve motivation-related behaviors or not, can also significantly influence hippocampal neural representations. For instance, in tasks where rewards are linked to specific locations, there may be an increased number of place fields that over-represent those locations[33], and changes in motivational states can lead to a shift in the population vector correlations[34]. Moreover, hippocampal neurons have been shown to encode reward-related goals or salience signals at the single-cell level. For instance, food-seeking behaviors can modulate the tuning and distribution of place fields that were previously detected during random foraging[16,35], and the firing rate of place cells may increase as animals approach a reward or encounter reward-related cues[11,13,36,37]. In the present study, mice were not rewarded based on

exploration behaviors, and the reward delivery site was located separately from the area where animals performed exploration. The significantly lower activity of oePCs to food rewards and the insensitivity of RA cells to LEC inhibition indicate that oePCs represent a distinct cell group from RA cells. These observations suggest that the hippocampus employs different neural units to encode reward-related information and exploratory intentions.

In humans and primates, hippocampal neurons are known to encode not only the parameters of physical space but also cognitive variables associated with non-spatial abstractions[38-40]. Similarly, rodents exhibit a range of behaviors that involve abstract thinking or intention. Recent research has demonstrated that the mouse hippocampus can encode abstract variables during navigation, incorporating these abstract variables into existing representations of physical variables[41]. As a result, internal variables, such as volitional navigation intentions and motivational importance previously shown to influence theta oscillations expression in the rat hippocampus[42], may also be represented as a form of abstract variable within neural manifolds. Much like place cells, hippocampal neurons encoding abstract values also exhibit remapping in novel contexts[39]. Our results have revealed substantial changes in oePC representations when the animals were transitioned to a novel maze. The question that arises is why the representation of both spatial information and the exploratory intentions of oePCs change in a novel context. One plausible explanation is that as contextual information is stored as independent representations for a multitude of environments, a change in a context likely reflects a hidden state inference of the animal's subjective judgement of the environment[43]. Each distinct hidden state can be encoded as an individual state space[44], with internal variables such as abstract values or navigation intentions being encoded in different dimensions together with other physical variables related to the context. Consequently, state space change leads to extensive alterations in the neural coding of all variables across dimensions within the hippocampus.

The medial entorhinal cortex provides an allocentric framework to the hippocampus[45], while the LEC is thought to provide egocentric information about discrete items or sensory cues[29]. Interestingly, LEC neurons can fire at one or multiple objects, but their activity at the same location is not always consistent across sessions[46]. One explanation for this effect is that LEC may act as a filter or gate on the upstream object-related information from the perirhinal cortex, allowing only attended experiences to enter the hippocampus[29]. Our findings indicate that inactivation of LEC impairs the exploration-associated activity in oePCs. We speculate that after receiving attended information from the LEC, the hippocampus may be involved in processing spatial attention into motor intention[47], the latter of which is commonly associated with motor preparation of voluntary actions. Inactivation of LEC might reduce attended signals from the object, but the general information of the object could still be input from the direct, albeit weak, connections between the perirhinal cortex and hippocampus[48]. Thus, neural representations of oePCs may be endowed with information processed through the entorhinal-hippocampal system, which includes not only external spatial/objective information but also internal attentional-intentional signals. These signals are likely an important component for a cognitive map that, beyond external spatial variables, may also include internal variables carrying mental processing-related information. These internal variables, such as animal's intentions independent of reward or environmental stimuli, could guide an individual's intentional navigation in the environment.

## Methods

### Animals
The animal experiments reported in this study adhered to the ARRIVE Guidelines and were conducted in accordance with the guidelines for the Care and Use of Experimental Animals at ShanghaiTech University, which were approved by the Institutional Animal Care and Use Committee. Male C57BL/6 J mice aged 8 to 16 weeks and weighing approximately 25 g at the beginning of the experiment were used. All mice were individually housed under a 12-h light/dark cycle with *ad libitum* access to water and maintained at a constant temperature (20–26 °C) and 40–60% humidity. All behavioral testing and recording were performed during the light phase.

### Stereotaxic surgery and adeno-associated virus (AAV) injection
Prior to surgery, mice were anesthetized with 1.5-2% isoflurane and secured in a stereotaxic instrument. A small craniotomy was performed, and then 500 nl AVV2/9-hSyn-GCaMP6f was injected into the right hippocampus (AP -1.9 mm, ML +1.3 mm, DV 1.4 mm) at a rate of 20 nL /min using a microinjection system connected to a glass pipette. After the injection was complete, the mouse's scalp was sutured, and it was allowed to recover from anesthesia. After the injection of AAV, all mice were allowed a recovery period of two weeks to ensure adequate expression of the viral vectors following surgery.

To suppress activity in both sides of the LEC, we utilized the Designer Receptor Exclusively Activated by Designer Drugs (DREADD) system, and injected 30 nL AAV2/9-hSyn-Cre in combination with AAV2/9-DIO-hM4Di-mCherry (at 1:1 ratio) or 30 nL AAV2/9-hSyn-mCherry to each side of LEC (AP -3.5 mm, ML ±4.1 mm, DV 4.35 mm). The hM4Di ligand Clozapine N-oxide (CNO) was dissolved in saline (0.9% NaCl solution) and further diluted to a working concentration of 5 mg/ml, which was stored at −80 °C. On the day of the experiment, each mouse was subjected to a behavioral test in either a Pre-saline or Pre-CNO session before being administered either saline or CNO. Approximately 30 min before the second session of behavioral tests, mice received an intraperitoneal (i.p.) injection of either saline or CNO (5 mg/kg).

### Gradient Index (GRIN) lens implantation
Implantation of GRIN lens was performed 2–3 weeks after virus injection. Prior to surgery, mice were injected with 20 mg/Kg Carprofen and 0.2 mg/Kg Dexamethasone to minimize issues of swelling and inflammation, 10 mg/Kg Enrofloxacin was given to prevent bacterial infection, and 0.5 ml saline was given to prevent dehydration. For 3 days following the surgery, the mice received the same dosage of Carprofen and enrofloxacin. Under anesthesia, a craniotomy (2 mm diameter) was performed above the viral injection site (centered at AP -1.9 mm, ML +1.3 mm). The cortical tissue above the targeted implantation site was carefully aspirated using 27-gauge blunt needles. After the corpus callosum was partially removed, and bleeding was stopped, a GRIN lens (1.8 mm diameter) was slowly inserted into the craniotomy and gently placed at the targeted site. Dental cement was used to seal and cover the exposed skull. The mouse was transferred to its home cage to recover from anesthesia and monitored until it was ambulatory. After the 2-week recovery period, the mice were anesthetized and the optimal field of view was identified using the miniscope, and the baseplate was fixed onto the skull with dental cement.

### Behavioral tests and miniscope imaging
We used two acrylic apparatuses, a square-shaped maze (30 cm length × 30 cm width × 30 cm height) or a circular-shaped maze (35 cm diameter × 30 cm height), for the behavioral maze tests. In both apparatuses, an acrylic cylinder (20 cm diameter × 20 cm height) was fixed concentrically to the maze to build annular tracks with about 5 cm width. The apparatus was resting on a table with multiple background cues, and each side of the inner walls was labelled with different cues. Both mazes were opaque and white.

Before training, mice were food-restricted and maintained at around 85% of their initial weight. Each mouse was habituated and trained in one of the two apparatuses (Supplementary Fig. 1). During the two-day habituation in the maze, mice were allowed to freely

explore the track, and a sugared-milk pellet was delivered through a reward port at a fixed position every time the mouse approached the reward position. In the training phase, mice were trained to run counter-clockwise along the track. The reward was delivered every time after mice completed a lap with the correct running direction, and an air puff was delivered when they ran in the opposite direction. Each mouse underwent a daily 20-min training session and reached an ~85% rate of correct-direction laps over all laps in approximately 10 days. During the last three days of training, different objects were placed at fixed positions at the 3-, 6-, or 9- o'clock relative to the reward site; and the animal was habituated to a sham miniscope for about 10 min each day.

On the first day of the test period, CA1 neurons were imaged, and oePCs were identified. If the animal's behavior did not meet the set criteria (a minimum of three exploration and three non-exploration bouts recorded for at least one object), additional recording sessions were scheduled for subsequent days as needed. After the identification of oePCs, further experiments were conducted on subsequent days, including assessments of cross-day stability, object displacement, object replacement, object removal, transitioning to novel maze, chemogenetic manipulations, and etc. Among these experiments, those involving transitioning to novel maze and chemogenetic manipulations were conducted with two imaging sessions daily, with a 30-min inter-session interval. Notably, to prevent potential alterations in the properties of already identified oePCs due to behavioral manipulation, each of the aforementioned experiments was conducted with separate groups of animals in most cases.

In the test sessions, we used the open-source miniscope system (V3) for in vivo Ca$^{2+}$ imaging and behavioral recording. The head-mounted scope has a mass of about 3 g and uses a single, flexible coaxial cable to carry power, control signals and imaging data to custom open-source data acquisition (DAQ) hardware and software[49,50]. Mice were handled and habituated to the miniscope for about three minutes before each imaging session. The apparatus, cues, and objects used in the first imaging sessions were consistent with those in training sessions unless specifically stated. The position of the animal was captured simultaneously with Ca$^{2+}$ imaging using an over-head behavioral camera (30 frames per second) with the MiniScope-Control program (https://github.com/daharoni/Miniscope_DAQ_Software).

## Position tracking and behavioral analysis

The position and speed of the animal was extracted from the behavior videos using modified code from the MiniscopeAnalysis package, which can be found at https://github.com/daharoni/Miniscope_Analysis. The frames from the behavioral videos were synchronized offline with the Ca$^{2+}$ imaging data. The cartesian coordinates of the mouse's position were then converted into polar coordinates, using the center of the square or circular maze as the reference point. For computing spatial information, the annular track was divided into 24 bins in a counter-clockwise direction, with each bin covering approximately 3 cm distance.

Behavioral bouts were defined as the period from when the mouse entered the $1/4\pi$ (in radians) in the track prior to the object until the $1/4\pi$ passing the object. The exploration and non-exploration behaviors were meticulously annotated by blinded, trained observers on a frame-by-frame basis. It was characterized by instances where mice engaged in sniffing, whisking, or touching the object with their nose or forepaws, or when they ambulated directly towards the object. Non-exploration was characterized by instances where mice traversed past the object without exhibiting discernible interaction with the object. The validity of the classification of the annotated behavioral bouts was subsequently confirmed by evaluating the difference in behavioral variables and the speed-time relationship from each bin (Fig. 1e, f and Supplementary Fig. 1).

Behavioral bouts with ambiguous classification were excluded from further analysis. Sequencing of exploration and non-exploration laps was depicted in Supplementary Fig. 13.

## Pre-processing of Ca$^{2+}$ imaging data

Pre-processing of Ca$^{2+}$ imaging data was performed using the MiniscopeAnalysis pipeline with slight modifications. Initially, the Ca$^{2+}$ imaging data was subjected to motion correction using a non-rigid motion correction algorithm (NoRMCorre)[51]. Next, the constrained nonnegative matrix factorization for microendoscope data (CNMF-E) algorithm was applied to obtain the spatial footprints (SFP) of individual neurons and remove the noise and background fluctuations from the raw trace, resulting in the temporal fluorescence intensity curves of each neuron[52]. The inferred neuronal activity of observed cells was reconstructed from the temporal fluorescence intensity curves using the algorithm for calibrated spike inference of Ca$^{2+}$ data using deep networks (CASCADE)[53]. Because CNMF-E uses complex methods for background subtraction, the neuronal activity was computed using the raw output of inferred firing probability by CASCADE multiplied by the frame rate and was reported in arbitrary units (arb. units) instead of absolute spike rates (https://github.com/HelmchenLabSoftware/Cascade).

The same neurons that were activated across two sessions were identified by the SFP correlation using the cell registration method[54]. The method used SFP from the early-sessions as a reference map, and aligned with this map the SFP from the post-sessions after correction of position offset and rotation. Then the method models the distribution of centroid distances for neighboring cells from early and post sessions and gets their weighted sum to determine whether they are the same cells or different cells. The method provides a P$_{same}$ registration threshold that is optimized to the dataset of each mouse. In our experiments, a pair of cells was considered to have the same identity if P$_{same}$ >=0.5. The centroid distance between a pair of cells which had the same identity was less than 12 μm. All tracked ROIs from pairs SPF images were plotted and manually inspected for quality.

## Spatial information and place cells

We computed the spatial information (SI) of each cell was computed using the synchronized position of the mice and the activity-time vectors using the reconstructed spike rates. To compute the spatial information and cPC/oePC identities, we used only completed laps during which the animal started from the reward site, ran through the annular track with correct direction, and return to the same site before retrieving the reward pallet. The periods of the food consumption were excluded from the analysis. The annular track was divided into 24 bins in an anti-clockwise direction starting from the reward site to generate a position-activity vector (laps × bins) for each cell (Fig. 1C bottom). We calculated SI of each cell was computed as mutual information (in bits) using the formula:

$$SI = \sum_{i}^{bins} p_i(r_i/\bar{r})\log_2(r_i/\bar{r}) \tag{1}$$

where $i$ is the spatial bin number, $p_i$ is the probability for occupancy of bin $i$, $r_i$ is the mean inferred activity for bin $i$, and $\bar{r}$ is the overall mean activity. The position-activity vector of each cell was shifted by a random time offset and the SI was calculated as a chance SI. We repeated this procedure of random shifts 100 times for each cell to obtain a distribution of chance SI. The significance of the observed SI was determined by converting it into a z-score based on the distribution of chance SI values of the same cell. We considered a neuron with the significance of SI ≥ 1.65, which corresponds to a probability of less than 5% of the chance occurrence, as a place cell.

To calculate the Center of Mass (COM) of place fields, a polygon was generated for each cell using the averaged bin activity and the

radial coordinates in response to each angular bin, and the centroid of the polygon was computed. The COM was determined using the angular coordinates of the centroid. We did not provide any special treatment to cells that might have multiple fields. The change of COM (ΔCOM) was calculated as the absolute difference between the COMs of two conditions for a cell. A place field was determined as a continuous region consisting of bins in which the activity of each bin exceed 20% of the maximum activity of the cell.

## Identification and characteristics of oePCs

The oePCs were identified from test sessions, each consisting of a minimum of three exploration and three non-exploration bouts recorded for at least one of the objects. Each behavioral bout covered 1/2 π of the track, spanning 1/4π prior to the object to 1/4π after the object. For each object, the maximum binned activity was computed for each exploration and non-exploration bouts, and the difference in mean between the two behaviors was determined. To assess significance was determined using bootstrap methods, in which the identity of exploration/non-exploration bouts were randomly shuffled, and the difference in mean was computed as chance data. This shuffling was repeated 1,000 times to establish the likelihood that the observed difference in mean could have emerged by chance. A cell was defined as an oePC if two conditions are satisfied: the observed difference in mean activity exceeded 99.0% of the chance, and the significance of SI ≥ 1.65 during the exploration laps. Cells with a maximum inferred activity lower than 2 arb. units were excluded.

To show the lap-average activity of oePCs or cPCs in exploration and non-exploration laps, we computed the bin-activity vector by averaging the activity across exploration or non-exploration laps and normalizing the data to the maximum activity in the cell's exploration vector. A difference-activity vector was computed by subtracting the bin-activity vector for non-exploration from that for exploration. To quantify the properties of oePC and cPC, several indices were computed from the bin-activity vectors.

The difference index was computed using the formula:

$$\text{Difference index} = \frac{\max(r_{\text{exp}}) - \max(r_{nonexp})}{\max(r_{\text{exp}}) + \max(r_{nonexp})} \tag{2}$$

Where $r_{\text{exp}}$ and $r_{nonexp}$ are the values from the bin-activity vector for exploration and non-exploration, respectively, in response to the three bins with the object bin as the center.

Spatial correlation was determined by the Pearson's correlation coefficient for the bin-activity vector of each registered cell in a pair of sessions. The chance levels of spatial correlation were determined by calculating the Pearson's correlation coefficient after the bin-activity vectors were shifted with a random time offset.

Reward-associated cells were identified if the cell's activity in the bins proximal to the reward location exceeded 2 standard deviations of the activity in all bins.

To compare the neuronal activity during baseline, exploration, reward, or other event-related periods, we computed the frame-by-frame speed vector using modified code from the MiniscopeAnalysis package. The exploration and reward periods were annotated as the speed-drop periods prior to object exploration or reward. The reward periods were annotated as mice approached the food reward, and the periods of the food consumption were excluded from the analysis. The other events were annotated as speed-drop periods at locations outside of the object exploration-associated place fields or reward sites. The baseline activity was calculated using the activity from all time points within laps, excluding those during exploration, reward, and other periods. The relationship between speed and activity was determined by Pearson's correlation coefficients. Statistical significance for the negative correlation between speed and activity for

each cell was assessed by comparing the observed correlation coefficient to the 95th percentile of the correlation coefficients obtained from shuffled datasets.

## Decoding

To decode animal behavior from neural activity, we employed a support vector machine (SVM) classifier with linear kernels. Specifically, we aimed to determine whether the neural activity of oePCs could reliably encode exploration versus non-exploration behaviors at different position bins. To do this, we labeled each behavioral bout as either 1 for exploration or −1 for nonexploration, and used these labels as our response variables. The oePC activity from the position-activity vector at the corresponding bin was then used as predictor variables for decoding the behavior. We evaluated the decoder performance using leave-one-out procedure, with each behavioral bout held out once.

In addition to examine the relationship between oePC activity and exploration behaviors, we also investigated whether the exploration/non-exploration behaviors were influenced by the animal's radial positions. To do this, we used a linear SVM classifier and the mean radial positions (Rho) at the corresponding bin as the predictor variables.

Prediction accuracy was determined by calculating the proportion of success that are made:

$$\frac{1}{k}\sum_{i=1}^{k} I(y_i = \hat{y}_i) \tag{3}$$

where $y_i$ is the $i$th observation in the response variables, $\hat{y}_i$ is the predicted class label for the $i$th observation, and $I(y_i = \hat{y}_i)$ equals 1 if $y_i = \hat{y}_i$ and zero if $y_i \neq \hat{y}_i$. The chance level of decoding performance was computed using randomly permuted period labels for the response variables. Prediction accuracy was considered statistically significant if it exceeded the 95th percentile of the distribution of chance accuracy from 100× surrogate data, corresponding to a percentile of <5% of chance (P value).

## Identification of Off-track positions and outlier dead direction points

Data points corresponding to off-track positions were determined on a bin-by-bin basis. Using data points from pooled non-exploration bouts within each bin, the mean and standard deviation (SD) of the radial distance (Rho) were first calculated for each bin. Threshold values for each bin were then established at mean ± 2 × SD. Rho values from the exploration bouts were subsequently analyzed, and any data that fell outside the threshold values of its corresponding bin was considered off-track.

A similar approach was used to identify outlier head direction points. Using data points from pooled non-exploration bouts within each bin, the mean and SD of the head direction angles were calculated for each bin. Threshold values for each bin were established at mean ± 2 × SD. Head direction angles from the exploration bouts were subsequently analyzed, and any data outside the threshold values of its corresponding bin were considered an outlier.

Following the exclusion of off-track or outlier head direction points, a two-sided Mann-Whitney U test was performed to compare neuronal activity between exploration and non-exploration bouts. A statistical significance level of ≤0.05 was applied to reject the hypothesis that the neuronal activity was similar between exploration and non-exploration. Cells that did not meet this statistical significance were excluded for subsequent analyses.

## Brain slice electrophysiology

Adult male C57BL/6 J mice (6−8 weeks old) were injected with combined AAV2/9-hSyn-Cre/ AAV2/9-DIO-hM4Di-mCherry or AAV2/9-

hSyn- mCherry in the LEC. Two weeks later, they were anesthetized with tribromoethanol (100 mg/kg, intraperitoneal injection) and then transcardially perfused with ice-cold, oxygenated (95% $O_2$, 5% $CO_2$) N-methyl-D-glucamine solution. This solution included 93 mM N-methyl-D-glucamine, 93 mM hydrogen chloride, 2.5 mM potassium chloride, 1.25 mM monosodium phosphate, 10 mM magnesium sulphate, 30 mM sodium bicarbonate, 25 mM glucose, 20 mM HEPES, 5 mM sodium ascorbate, 3 mM sodium pyruvate, and 2 mM thiourea. After perfusion, the brain was quickly dissected out and immediately transferred into an ice-cold, oxygenated N-methyl-D-glucamine artificial cerebrospinal fluid solution. Subsequently, we sectioned brain tissue coronally at a thickness of 300-μm in the same buffer using a vibratome (VT1200 S, Leica, Germany). The brain slices containing the LEC were incubated in oxygenated N-methyl-D-glucamine artificial cerebrospinal fluid at 32 °C for 10–15 min, then transferred to a normal oxygenated solution of artificial cerebrospinal fluid (126 mM sodium chloride, 2.5 mM potassium chloride, 1.25 mM monosodium phosphate, 2 mM magnesium sulphate, 10 mM glucose, 26 mM sodium bicarbonate, 2 mM calcium chloride) at room temperature for 1 h. All chemicals used in slice preparation were purchased from Sigma-Aldrich (St. Louis, MO, USA).

Slices were transferred to the recording chamber that was submerged and superfused with artificial cerebrospinal fluid at a rate of 2–3 mL/min at 28 °C. Whole-cell patch-clamp recordings were made from LEC neurons visualized with an Olympus BX61W1 microscope (equipped with mCherry filters) using infrared video microscopy and differential interference contrast optics. Patch electrodes (3–5 MΩ) were pulled with a pipette puller (P2000, Sutter Instrument; USA) from borosilicate glass capillaries. Whole-cell recordings were obtained with an internal solution containing (in mM): 135 mM potassium gluconate, 5 mM sodium chloride, 10 mM HEPES, 1 mM EGTA, 0.3 mM sodium guanosine 5′-triphosphate sodium salt hydrate, 2 mM magnesium adenosine triphosphate, 1 mM magnesium chloride (280 to 300 mOsm; pH 7.2).

To assess the effects of pharmacogenetic inhibition of LEC neurons, CNO (10 μM) was added to the artificial cerebrospinal fluid perfusion and a bath applied to the recorded neurons. The resting membrane potential of a neuron was obtained under the current clamp (I = 0 pA). For action potentials evoked by current injections, a current-step protocol (from −20 to +360 pA, with 20 pA increment for recording and the step currents during the recording of the action potential is 400 ms) was run and repeated. Electrophysiological recordings were acquired with a MultiClamp 700B amplifier (Molecular Devices) and Clampex 10.6 software. Signals were low-pass filtered at 2 kHz and digitized at 10 kHz using Digidata 1550B (Molecular Devices). Recordings with Rs >30 MΩ were excluded from statistical analysis. Offline electrophysiological data analysis was performed with Clampfit 10.6 (Molecular Devices).

### Reporting summary
Further information on research design is available in the Nature Portfolio Reporting Summary linked to this article.

## Data availability
Data generated in this study are provided in the Source Data file. Source data are provided with this paper.

## Code availability
The custom code that supports the findings of this study is available at GitHub and can be accessed via https://github.com/ZhouNinglab/Conjunctive-encoding-of-exploratory-intentions-and-spatial-information-in-the-hippocampus. Any additional information will be available from the authors upon request.

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

## Acknowledgements
We thank Dr. Wan-Chen Jiang (Professor at SLST, ShanghaiTech University) for insightful feedback on the experiment design, Xueying Fang for help with cell registration, Yu-Ying Mei for help with miniscope techniques, and Zhen-Zhen Wu for help with behavioral experiments. This work was supported by ShanghaiTech University, National Science Foundation of China (32170959) to N. Z., National Science and Technology Innovation (2030 Brain Science and Brain-Like Intelligence Technology Major Project 2021ZD0203900) to X.-N.Z., National Science and Technology Council (MOST110-2320-B-039-010-MY3, MOST111-2321-B-A49-005, NSTC 112-2320-B-039-056, NSTC 112-2321-B-A49-009) and in part by China Medical University (CMU109-MF-31) to D.C.W.

## Author contributions
Conceptualization: Y.F.Z., D.C.W., N.Z.; Methodology: Y.F.Z., D.C.W., N.Z.; Investigation: Y.F.Z., K.X.Y., Y.L.C., X.N.Z., R.L., H.Z.; Formal analysis: Y.F.Z., Y.L.C., K.X.Y.; Funding acquisition: X.N.Z., D.C.W., N.Z.; Project administration: J.H., R.C.S., N.Z.; Supervision: X.N.Z., D.C.W., N.Z.; Writing: Original Draft: Y.F.Z., N.Z.: Writing – review & editing: D.C.W., N.Z.

## Competing interests
The authors declare no competing interests.
