## [Peer Review File · Nature Communications]

Conjunctive Encoding of Exploratory Intentions and Spatial Information in the HippocampusREVIEWER COMMENTS

Reviewer #1 (Remarks to the Author):

In this manuscript the authors examined the firing of CA1 neurons in a spatial task with calcium imaging and reported a few interesting findings. Firstly, some CA1 cells were active in their place fields only when the animals actively explored an object. Secondly, the firing of these cells appeared to be independent of the objects to be explored, the rewards, or getting away from the regular locomotion trajectories. And thirdly, LEC appeared to have minor impact on this type of firing. Consistent with a few recent studies, this manuscript demonstrates that the CA1 neurons integrate spatial representation and non-spatial information. The work will likely attract broad audience in the field of neuroscience, especially researchers studying basic neuronal mechanisms of cognitive processing. The experiments were well designed and carefully carried out. The data were clearly presented. I only have a few suggestions for the authors to further improve this manuscript.

1) Possibility of interpreting the data within the conceptual framework of the “spatial tuning” of CA1 pyramidal cells. Although called “place cells”, place cells’ firing is not solely determined by locations. For example, in the same location in a linear track, different place cells are activated when animals travel “up” in the track compared to those activated when animals travel “down” the track. In the circular track used in this study, the mice were primarily traveling in a counterclockwise direction. When they encountered objects, the mice may change head orientation to some degree and activate place cells that were supposedly to be activated when the mice were not traveling in counterclockwise directions.

2) Evidence of encoding of “intention”. Although the authors have carefully excluded many possibilities for explaining the place cell firing related to explorative behavior, it may still be too early to conclude that this firing encodes “intentions” to explore. Since the appearance of these neuronal activities relies on the presence of objects to be explored, it may not be possible to completely exclude the possibility that these activates represent something related to the objects. For example, since these neuronal activities increased before actual exploration, these activities may reflect the presence of an object, but not the identities or detailed features of the objects. Therefore changing objects may not solve this issue completely. Related to this, the authors may want to include a more detailed description of the objects used.

3) The chemogenetic inhibition of LEC showed minimal impacts on the CA1 neuronal activity, which could be due to a limited role of LEC in this type of neuronal activities or due to the low effectiveness of hM4Di in LEC. Slice physiology may help to address this issue.

4) The inhibitory interneurons, though a small fraction in the CA1 pyramidal layer, may still contribute to the calcium signal analyzed here. The authors may want to discuss this issue.

5) The paper by Nieh EH et al., (Nature. 2021 Jul;595(7865):80-84.) provide strong support for this current study. The authors may want to integrate it into the discussion.

Reviewer #2 (Remarks to the Author):

The study examines how place cells respond while animals traverse a track that contains object cues and a reward at the end. It presents a large sample of place cells recorded using calcium imaging and specifically examines how these cells respond to periods of exploration around the objects. These periods are termed voluntary exploration.

The data are convincing in showing that place cells encode exploratory behaviour differently to non-exploratory behaviour in the same spatial location. This type of change to place cell activity as a result of the change to behavioral context within a consistent spatial location has been shown many times before with the classic example being the splitter cells from Wood et al. 2000. Since then, lots of studies have shown that place cells will encode different behavioral contexts (potentially intentions) in the same spatial location. For example, the Dudchenko lab has shown that different place cells will encode the intended destination of an animal at the start of the maze that has multiple different potential destinations.

In the current experiment the authors suggest that behaviors / cellular activity based on voluntary intentions could be dissociated from behaviors / cellular activity based on rewards. They claim that their findings support a new type of code for voluntary intention that is different to splitter type activity and independent of reward. However, animals choosing to run a particular route to a reward are demonstrating a voluntary intention. It could also be argued that exploration of an interesting object is rewarding. This presents 2 issues: firstly that it isn't clear why rewarded vs. non-rewarded intentions would be coded differently and secondly what is the nature of reward? While I think the current study presents clear and convincing evidence of place cell modulation by behavioral context, I don't think it provides evidence for reward independent voluntary behavior. Indeed, I don't think such a class of behavior would be likely.

In the example at the beginning of the paper voluntary intention is illustrated by a situation where we stop to get a newspaper. However, the newspaper in this example is a reward. In the current task the animals stop to explore an object and so a better metaphor might be stopping to look at a statue. Again

though - are we sure this isn't rewarding? Even if we accept that it isn't why are reward based behaviors not driven by voluntary intention? These 2 issues need to be explained.

Specific issues to address in the paper:

Please illustrate the 24 angular bins in 1C. This might be the quality of the image but it looks like the diagrams have more than 24 bins?

Some aspects of the analysis are a little circular. Exploratory bouts are scored manually as periods where the animal slows to explore the object. Figures 1E and 1F show that 'intentional exploration' is slower than non-exploration and that this demonstrates clear differences between the 2 types of behaviour. However, non-exploration is defined as animals showing "no apparent slowing" on line 557 and so the data in 1 E & F represent only that the analysis has worked, not that there are 2 classes of behaviour. Similarly, intentional exploratory bouts are defined by their proximity to objects (lines 553-554) and Figures 2D&E just show that this method worked, not that intentional behaviour is grouped around objects as suggested here: "Compared to cPCs, whose place fields were evenly distributed across the entire track, the place fields of iePCs were maximally distributed with proximity to the locations of objects, as shown in the activity map sorted according to place field centers (Fig. 2D, E) and distribution of the center of mass (COM) (Fig. 2F)". The analysis only permits intentional behaviour to be found around objects.

It would be interesting to illustrate the 2SD in rho on 3A/B.

The authors propose that iePCs and reward cells are fundamentally different classes of cells. The current setup doesn't allow this distinction to be made. They show clearly that cells that respond to the reward location do not respond to the object locations. However, Figure 1E shows that iePCs typically only respond to one of the object locations. Using the same logic you could say that iePCs that respond at object position 1 are fundamentally different to all other cells recorded. To demonstrate that these cells are not reward cells the reward would need to be presented in the object locations and a lack of activity would need to be demonstrated. At present the data show clearly that populations of place cells respond to all of the interesting areas in the arena and that each interesting area is represented by different place cells. However, they do not show that rewards are represented differently to other interesting areas.

The most convincing argument that the iePCs are object oriented is the fact that their activity is disrupted by inhibiting LEC input. It would be informative to see whether reward cells are also inhibited in the LEC condition. If they are not this would be good evidence that iePCs are not reward related.

Reviewer #3 (Remarks to the Author):

In this study, the authors employed miniature endoscope imaging to record the activity of a population of dorsal CA1 neurons during an intentional exploration task. They found that a small population of CA1 neurons not only encode for the spatial information but also the animal's investigatory intention. Furthermore, inhibiting the LEC maintained the spatial properties of these neurons but reduced the activity between exploration and non-exploration intention. Together, the authors suggest that this newly identified group of CA1 neurons could be critical in bridging external information (place fields) with internal intention (exploration).

The study is definitely interesting and novel, and the results will bring a new perspective to the area of hippocampal place field. Many control parameters and different maze settings were conducted to test the properties of these iePCs, which is a lot of work considering each experiment is combined with in vivo calcium imaging. However, more detail about the behavior task and the analyses should be provided. In addition, a lot of the figures were not labelled, which makes the manuscript a bit difficult to read/understand. Below I listed the questions that I had.

Major:

1. What's the purpose of the square- and circular-shaped maze? It seems like most of the analyzes were done in the square maze or did the authors combine all the trials?
2. It would be nice if the authors could describe the behavioral paradigm in detail.
 - In the methods, it only says 'animals reached ~85% rate of correct-direction laps over all laps in about 10 days, etc....'. How many laps does the animal run on the imaging days.
 - '1-3 different objects were placed'  how many objects were placed exactly? If more than one, is the neuronal response for each object combined for the analyses?
 - 'Imaging sessions were performed every day for 1-2 sessions with 20min ITI'  how many days were imaged? For the 2 sessions within a day, was the object placed at the same location? If the animals are familiar with the object and maze, does that decrease their exploration trials? If yes, does the activity change between session 1 and session 2 within the same day?
 - What location was the reward given? Was it at a random location and how far away was it from the 3 objects?
 - All of these factors could affect the authors' interpretation on 'exploration' and 'intention'.

3. While it is nice the authors showed the location of iePCs in each animal in SFig 2, the image quality that I got from the PDF file is not very good. Can the authors provide a summary table of how many iePCs are identified in each animal? Moreover, do the authors observe a trend on whether these cells have a specific location in CA1 or are they intermingled among other place cells?

4. I'm confused about this 'off-track' position. It looks like the authors can better decode the behavior or iePC activity when they included off-track trials, so why do they keep saying they are 'excluding' them (line 125)?

5. Figure 5A – I presume this is done on expert animals? Did the authors follow the same cells and track their place fields when the object was moved away? Also, how many trials were performed for each new distance?

6. Figure 6B, how many exploration trials are in early vs. late? Do they have similar numbers, or do the early sessions have a lot more exploration trials?

7. Figure 6D, does replacing the max box induce new iePCs within the imaging field? It doesn't seem like it does, judging from Fig 6C, but why is that?

Minor:

1. Figure 2A, B – missing lap #

2. Figure 2D, E – the heat bar doesn't have a unit.

3. Figure 2F – there are no y-axis title for the spatial information and difference index.

4. Figure 4A should be shown like Figure 3D because it'll provide speed information as well. And it's missing the y-axis unit.

5. Is firing rate the same as spike rate (mentioned in figure 1)? And why is it 'au'?

6. Figure 6A and 6C and in general, more labels on the figures will help the readers to understand the plots easier without having to read the figure legends every single time. What are the two heat plots in the middle?

Reviewer #4 (Remarks to the Author):

This paper explores the intriguing possibility that the hippocampus contains neurons that respond to the animals' investigatory intentions. The authors use calcium imaging to record from area CA1 in the mouse during object exploratory and non-exploratory laps. The main finding is that a percentage of neurons in CA1 becomes active before exploratory behavior at specific locations and display significant reductions in activity during non-exploratory laps (the authors call these cells intentional exploration-dependent place cells, iePC). The authors show that the majority of iePC cells are not the same as those responding to reward location or object identity. For example, the cells display similar levels of activity when the objects are replaced in the same locations and, on average, the cells did not show increased activity after an object was removed (shown only in 9 cells, 3 mice, Figure S3). Finally, the authors show that inhibition of the lateral entorhinal cortex, diminishes the activity mismatch between exploratory and non-exploratory laps. Although the authors propose an interesting idea, the data do not support the claim that these cells reflect "voluntary intentions neurons". Alternative explanations need to be ruled out. Below I summarize the main points that need to be addressed to clarify the meaning and impact of the findings.

1. The use of the term "voluntary intentions" for the interpretation of rodent behavior is an overreach. The definition of voluntary intentions can be extremely subjective and, as pointed out in the literature, subject to philosophical biases (Libet, 1985). Voluntary intentions have been proposed not only to require endogenous responses, but also reflect a level of introspective awareness. This manuscript does not offer compelling evidence that mice have a level of cognitive abstraction for this process. Therefore, calling hippocampal cells "object-exploratory cells" would be a more cautious and appropriate way to represent the findings.
2. Along these lines, the introduction states that voluntary intention cells could code "what we want to do" (line 24). Implying voluntary intention in the mouse is deceiving, not only because it is not known if mice are capable of voluntary intention (point above), but also because some alternatives have not been ruled out (see points below).
3. The authors claim that the iePC cells are not reward cells because these cells (at least most of them) are not the same neurons that fire at the reward locations, suggesting that their role is independent of reward properties. The authors fail to note that there are several kinds of reward and different cells may respond to distinct reward characteristics. A subpopulation of cells may respond to edible rewards, whereas another subset may respond to object exploration. Different hippocampal cells have been shown to code reward locations, action value of reward, anticipation of reward, etc. Therefore, it is possible that distinct subpopulations also represent edible vs. object exploration reward value.

4. Additionally, hippocampal cells have been shown to fire in anticipation of reward (Lee et al. 2006; Jarzabowski et al., 2022). Since the iePC cells increase activity before the mouse reaches the objects (line 140), a crucial analysis would be to compare how the same cells respond when mice are approaching the food reward (before they reach the reward location rather than at the reward location).

5. One of the main points of the paper is that iePC cells reflect voluntary intention because their activity increases only when animals slow down to explore the objects. It would be important to see the sequence of exploratory and non-exploratory laps in different mice. Do non-exploratory laps occur more often after animals have explored the objects substantially? Although the authors give 3 days of habituation, it is different to see the objects early on the first day of testing vs. at the end of the testing session. Are exploratory laps observed during the object habituation period (3 days prior imaging)? How do the authors rule out that the non-exploratory laps are not the result of habituation to the objects rather than the voluntary intention to omit exploration? Rodents running in mazes containing objects or performing object place tasks show decreased exploration for the objects and places that are familiar (Manns and Eichenbaum, 2009). Therefore, a distribution of non-exploratory laps over time of testing will be important.

6. It is not clear why some animals were tested with 1 object and other with 3. Did the authors do some analysis to compare animals trained with 1 vs. 3 objects? Are the iePC cells specific to one object or multiple objects in sessions with 3 objects? Are there more exploratory laps observed in animals trained with 3 objects? For example, in Figure 1A the square and cylinder schematic contain 3 objects; however, Figure 1E shows an example of an exploratory and non-exploratory cell with 1 object. How did the authors decide when to use 1 vs. 3 objects? Why the method was not consistent across animals?

7. The authors rule out that object identity/novelty could trigger the differential activity between exploratory and non-exploratory laps because, when objects are replaced, no differences were observed between familiar and replacement conditions. When were the new objects replaced during testing? How much time elapsed between a regular session and one with object replacements? How different were the objects (a picture will be useful)? I found this experiment interesting because the results do not show a change in overall activity. However, I wondered if this experiment was conducted early during testing or after the animals were extremely familiar with the objects. Without knowing the specifics, it is impossible to determine the relevance of these findings. It is quite established that animals will explore new objects more than familiar ones, therefore, the lack of neural differences in this experiment is intriguing and surprising.

8. The object elimination described in Figure S3 included only 9 cells in 3 mice. Considering that there is some variability in the results, why did the authors only select a subset of mice/cells?

9. It is well known that mice find some objects more interesting than others. In sessions with 3 objects, does object preference interact with iePC cell activity? Since preferred objects should generate more exploratory behaviors than non-preferred ones, it would be interesting to determine if the DI index is affected for preferred vs. unpreferred objects.

10. It is really confusing to know precisely what the timeline/design of the experiment was. The Caption of Figure 1A mentions 9 experiments. Does this include sessions in the square and the cylinder across days? How long was each session and how many days were recorded? The method states that after the period of object habituation (3 days), imaging sessions were performed every day for 1-2 sessions, but it is not clear how many days the testing involved.

11. The results should specify that the cells were tracked across sessions. The authors only clarify that neurons were tracked across sessions in the method. The fact that this explanation is not provided early on, leaves many points unclear. Are the 107 iePC cells the ones that have SFPs across sessions? Did the authors analyze iePC cells that were temporary active (cells that show exploratory activity only during one session but could not be tracked across days)? If so, are the temporary and consistently active iePC cells similar? How many cells were tracked relative to the total number of cells recorded? The free tracking algorithms sometimes capture artifacts, were some adjustments made to the code to make sure that the tracked cells were neurons?

12. I found the remapping observed after object displacement counterintuitive to the authors' argument. The results show that object displacement caused significant remapping and changes in the place fields' center of mass, with most fields disappearing when the object was moved 4 cm. If the iePC cells were indeed voluntary intention cells, then they should not have been affected by object displacement. Using the example provided by the authors, if someone wants to stop to buy the New York Times along his/her path to work, it should not matter if the location of an ambulatory seller changes from one day to another. The intention to stop and buy the newspaper should be the same regardless of the location of the seller. However, if the cells are merely coding object/place locations, then remapping makes a lot of sense.

13. The differential activity of iePC cells during exploratory vs. non-exploratory laps disappears when animals are placed in a new context (Figure 6D). Are the iePC cells normal place cells when animals are moved to a new box? If the subpopulation of iePC cells does not show the same properties across contexts, it would be important to discuss this characteristic in the discussion and provide a possible explanation.

14. The protocol to train and test the animals is confusing. According to the method, mice were first trained to run on tracks in either a square or a cylinder in one direction before starting recordings. Then, animals received 1 or 2 sessions (line 539) with the miniendoscope attached. Why some animals

received 1 and others 2 sessions? How long were the sessions? How did these 2 sessions generate 9 experiments mentioned in line 344? How many days the cell activity was recorded? Why were different groups of mice tested in the cylinder and square? Having a schematic of the training timeline and design in one of the Figures will be useful.

15. The authors refer to firing rate when describing calcium events (lines 41, 38, 588, 607, 612, 802, Figure S3, Figure S4 etc.). This is incorrect. Calcium imaging only provides an indirect measure of activity. I commend the authors for using CASCADE, but they should refer to “inferred spikes” or “inferred activity” not firing rate. Did the authors do anything to normalize fluorescence across cells and days? For example, do the authors use the raw output from CASCADE or do they normalize before or after?

16. To determine the significance of iePC cells the authors performed bootstrap analysis “the identity of exploration/non-exploration bouts were randomly shuffled, and the difference in mean was computed as chance data. This shuffling was repeated 10,000 times to establish the likelihood that the observed difference in mean could have emerged by chance”. 10,000 seems excessive. If 3 exploring and 3 non-exploring trials were included, there were 20 unique combinations.

17. For any given animal and session, how many laps were exploratory vs. non-exploratory? The authors report that sessions were included if at least there were 3 of each but the total number per animal would be useful.

18. The difference index was computed using the difference in max bin activity for exploration and non-exploration trials divided by their sum. The maximum activity was calculated over 5 bins with the object in the center. Considering that the arena was divided in 24 bins, each bin was 15 degrees ($360/24=15$). Therefore, the max activity was computed over 75 degrees. This seems quite excessive because with 3 objects the maximum activity will cover 225 degrees of the arena. What is the actual angular distance around each object? How much does the DI change if a more restricted radial measure is used in the analysis?

Minor comments

-I recommend thorough editing of the manuscript. Tenses are mixed in many paragraphs, including the abstract (second sentence mixes present and past tense)

-line 30 should be “as they travel through that location.”

-The Figure caption of Figure 3C says top and bottom and it should be left and right.

-The authors use probability and firing rate without defining these terms. Throughout the paper, firing rate should be replaced by inferred activity or inferred rate. Probability should be defined.

-the design indicates that animals are familiarized to the objects prior the imaging. When is the baseline activity recorded (line 154)? Neural activity when animals decrease speed in areas outside the object regions is compared with baseline, but it was not clear when the baseline was computed.

-in line 190 the term “control experiments” is misleading. These are control laps where even and uneven exploratory laps are correlated. It is not a separate experiment.

General response to Reviewers' comments

We express our gratitude to all reviewers for the invaluable feedback provided on our manuscript. We are pleased that the reviewers found our work interesting, and we highly appreciate their meticulous reading and many constructive thoughts and suggestions. In response to their comments, we have conducted a thorough set of experiments to address each comment systematically. We believe that our manuscript has been greatly improved. Below, we present a brief summary of how we have addressed the main concerns, followed by detailed responses to each comment.

All reviewers raised a general question regarding the evidence of encoding “voluntary intention” in mice. We acknowledge that mouse behaviors are primarily driven by instinct, learned associations, and responses to environmental cues, all linked to biological needs. These behaviors differ from what we typically attribute to “voluntary intentions” or a level of introspective awareness like those of humans. Therefore, exploration or investigatory behaviors in rodents represent a form of intended behavior rather than the kind of voluntary, intentional behaviors associated with conscious planning, abstract thinking, or complex cognitive processes. Accordingly, we have removed the terms “voluntary intentions” and “voluntary behaviors” from the manuscript in the revised version. Following the suggestion of Reviewer #4, we have also modified the previously used term “intentional exploration-dependent place cells (iePC)” to “object exploration-dependent place cells” (oePC) as a more cautious way to represent these findings. This term has been used in the revised manuscript and in the following context of this response letter.

Nevertheless, based on our observations, we maintain our assertion that oePCs genuinely represent exploratory intentions, distinct from neural representations in response to food rewards or the presence of objects. To reinforce this conclusion, we conducted a series of additional experiments. Here we briefly summarize the primary findings from the major experiments, while more detailed responses and figures provided after each relevant Reviewer's comment.

In the first experiment, we investigated whether oePCs remain active when an object is substituted with a food reward at the same location. During a recording session, we initiated the experiment by allowing the mice to freely explore the familiar object, following our original experimental design. This enabled the identification of oePCs from the object exploration and non-exploration laps. In a subsequent lap, we removed the existing object and introduced a food pallet at the exact same location before the mouse reached it. In the following laps, a reward was released at the same location after the mice arrived, and the neuronal activity in these laps was analyzed as responses to food rewards. Our results revealed that the previously identified oePCs displayed significantly reduced activity during the reward-expecting laps (Fig. 4g-j), indicating a distinct response of these cells to object exploration versus food anticipation.

In the second set of experiments, we have replicated the chemogenetic manipulation experiments of the LEC. Addressing the suggestion from Reviewer #1, we assessed the efficacy of hM4Di using path clamp recordings in brain slices (Supplementary Fig. 11) and chose a more potent batch of hM4Di AAV to examine the impact of LEC inhibition on the responses of both oePCs and reward cells. The inhibition of LEC with CNO significantly reduced the activity of oePCs in response to object exploration, while it had no significant effect on the activity of reward-associated (RA) cells during food anticipation in the same experiments (Fig. 7, Supplementary Fig. 12). These results collectively highlight a clear differentiation between oePCs and RA cells as distinct cell types.

In the third experiment, we introduced a new apparatus (Supplementary Fig. 6) to investigate whether oePCs respond to the presence of the object or the intention for exploration. We positioned the object outside of the track and obstruct its visibility using opaque partitions. After being trained in this behavioral arena and having well learned the location of the object, the mice were allowed to freely explore the concealed object by passing through a designated door or to bypass the door location without engaging in exploration. It is worth emphasizing that the mice were unable to visually perceive the object unless they chose to pass through the door and engage in exploration. Applying the same criteria for identification of oePCs in the manuscript, we proceeded to identify oePCs and uncover a group of neurons that specifically increase activity before the exploration of objects. Activity maps of these cells clearly show firing patterns prior to the mice reaching the designated door and before direct visualization of the presence of the object. Notably, these oePCs exhibit properties consistent with those identified in the initial experiments. These data collectively provide compelling evidence that oePCs encode the intention of object exploration rather than relying on the visual perception of the object's presence or properties.

Based on these results, we propose that oePCs conjunctively encode exploratory intentions and spatial information. While we acknowledge the inherent limitation in directly discerning rodents' true "intentions", we contend that the activity of oePCs is most accurately interpreted through the lens of exploratory intentions rather than anticipation to rewards, visual perception of objects, or alterations in physical variables such as location or head directions. We hope the Reviewers will agree that the additional experiments sufficiently support this conclusion.

Furthermore, we have undertaken substantial revisions to the Method section to improve clarity regarding the experimental setup and timeline. We have expanded the sample size for certain experiments to bolster the robustness of our findings. Some data and figures have been updated according to increased sample size or modified analysis. All data and statistical analysis have been included in the Source Data file. In addition, we have refined the discussion to address the relationship between our findings and recent discoveries reported by other research groups concerning neural encoding of internal abstract variables in the hippocampus. We believe that these additional results and improved presentation will address all Reviewers' concerns.

Point-to-point responses to Reviewer comments

Reviewer #1 (Remarks to the Author):

In this manuscript the authors examined the firing of CA1 neurons in a spatial task with calcium imaging and reported a few interesting findings. Firstly, some CA1 cells were active in their place fields only when the animals actively explored an object. Secondly, the firing of these cells appeared to be independent of the objects to be explored, the rewards, or getting away from the regular locomotion trajectories. And thirdly, LEC appeared to have minor impact on this type of firing. Consistent with a few recent studies, this manuscript demonstrates that the CA1 neurons integrate spatial representation and non-spatial information. The work will likely attract broad audience in the field of neuroscience, especially researchers studying basic neuronal mechanisms of cognitive processing. The experiments were well designed and carefully carried out. The data were clearly presented. I only have a few suggestions for the authors to further improve this manuscript.

Response: We thank the reviewer for the supportive comments.

1) Possibility of interpreting the data within the conceptual framework of the “spatial tuning” of CA1 pyramidal cells. Although called “place cells”, place cells’ firing is not solely determined by locations. For example, in the same location in a linear track, different place cells are activated when animals travel “up” in the track compared to those activated when animals travel “down” the track. In the circular track used in this study, the mice were primarily traveling in a counterclockwise direction. When they encountered objects, the mice may change head orientation to some degree and activate place cells that were supposedly to be activated when the mice were not traveling in counterclockwise directions.

Response: We appreciate the reviewer for raising this question. As evident from the firing map below (color bar) illustrating the activity of a representative oePC in relation to head direction (arrows) at various time points along the moving trajectory, it is clear that this cell’s activity is not solely dependent on head directions. To conduct a more in-depth analysis, we proceeded with a quantitative approach. We segmented exploration and non-exploration bouts into angular bins, calculating the mean and standard deviation of head direction angles within each bin based on all non-exploration bouts. During the exploration bouts, mice may exhibit some degree of variation in head direction, and these instances of “outlier head direction” were identified using the mean \pm 2SD for each bin. Following the exclusion of these “outlier head direction” points (illustrated as gray arrows in Supplementary Fig. 5), we conducted a two-sided Mann-Whitney U test to compare neuronal activity between exploration and non-exploration bouts. A statistical significance level of ≤ 0.05 was applied to reject the hypothesis that the neuronal activity remained similar if the mouse displayed comparable head directions at each angular position (See Methods, Line 775-784, page 25). By using this approach, we identified that 6 out of the initially identified 10 oePCs, with 4 cells from the square maze and 2 cells from the circular maze, did not meet this statistical significance, indicating that they might be modulated by head directions. These six cells have thus been excluded from the oePC category in the subsequent analysis.

Supplementary Figure 5. Firing maps in relation to head directions. The firing maps illustrate the inferred activity of a representative oePC in relation to head direction (indicated by arrows) at various time points along the moving trajectory during three non-exploratory (a) and exploratory bouts (b). (c) Same as in (b) for the same cell but after exclusion of outlier head direction points that are illustrated in gray. The right panels show the head-direction tuning curves under the corresponding conditions.

2) Evidence of encoding of “intention”. Although the authors have carefully excluded many possibilities for explaining the place cell firing related to explorative behavior, it may still be too early to conclude that this firing encodes “intentions” to explore. Since the appearance of these neuronal activities relies on the presence of objects to be explored, it may not be possible to completely exclude the possibility that these activities represent something related to the objects. For example, since these neuronal activities increased before actual exploration, these activities may reflect the presence of an object, but not the identities or detailed features of the objects. Therefore changing objects may not solve this issue completely. Related to this, the authors may want to include a more detailed description of the objects used.

Response: We agree with the reviewer that our initial experimental design did not provide a comprehensive insight into whether oePCs encoded “intentions” or merely “the presence of the objects”. In response to this important question, we designed a set of supplementary experiments. In these trials, we positioned the object outside of the track and obstructed its visibility using opaque partitions. After being trained in this behavioral arena and having well learned the location of the object, the mice were allowed to freely explore the concealed object by passing through a designated door or to bypass the door location without

engaging in exploration (Supplementary Fig. 6a). It is worth emphasizing that the mice were unable to visually perceive the object unless they chose to pass through the door and engaged in exploration.

Next, we excluded the data corresponding to the “off-track” points, which in this experiment denote the points where the mice passed through the door and went outside of the track (Supplementary Fig. 6b). Employing the same criteria for identification of oePCs in the manuscript, we proceeded to identify oePCs and uncover a group of neurons that specifically increase activity before exploration of the objects. Their activity maps clearly show that these cells fired prior to the mice reaching the designated door and before they directly visualized the presence of the object (Supplementary Fig. 6c). These oePCs exhibit properties consistent with those identified in the initial experiments (Supplementary Fig. 6d-f). These data collectively provide evidence that oePCs encode the intention of object exploration rather than visual perception of the object’s presence.

Supplementary Figure 6. Encoding of exploratory intentions by oePCs for indirectly visible objects. (a)

The experimental design diagram illustrating the object is positioned outside of the track with visibility obstructed by opaque partitions. Following training in this behavioral arena with objects, mice freely explored the concealed object by passing through a designated door or to bypass the door location without engaging in exploration. (b) Top view of the maze displaying the location of a sample object and rewards (R). The dashed curves represent the mouse’s trajectory during an exploration lap, with the red segment indicating “off-track” positions. (c) Color-coded neuronal activity of an example oePCs along the mouse’s trajectory, with each panel illustrating a lap during exploration (top) or non-exploration (bottom). These activity maps demonstrate increased oePC activity exclusively during exploration bouts, even when the object’s location was well-learned but not directly visible. (d) Lap-average activity of oePCs in exploration and non-exploration laps across all mice, sorted by place field centers. Spike rates were normalized to the peak activity of each neuron and sorted according to angular position. The right panel shows difference between the activity map in exploration and non-exploration laps. (e) The spatial index during exploratory laps (0.72 (0.50 – 0.88)) were significantly higher than non-exploratory laps (0.56 (0.26 – 0.7)) in identified oePCs (n = 22). (f) The difference index of oePC (0.89 (0.81 – 0.95)) was significantly higher than that of classic PCs (0.02 (-0.25 0.27), n = 310).

3) The chemogenetic inhibition of LEC showed minimal impacts on the CA1 neuronal activity, which could be due to a limited role of LEC in this type of neuronal activities or due to the low effectiveness of hM4Di in LEC. Slice physiology may help to address this issue.

Response: We thank the reviewer for raising this important issue. To address this concern, we conducted electrophysiological experiments using the original batch hM4Di AAVs, which were used in the initial experiments, and subjected them to patch clamp recordings in LEC slices. Surprisingly, we observed a notably low effectiveness with this batch. Consequently, we tested hM4Di AAVs sourced from different vendors, incorporating electrophysiological techniques. Ultimately, we adopted an approach that involved co-infusing AAV2/9-hsyn-Cre and AAV2/9-DIO-hM4Di-mCherry for chemogenetic inhibition. We have since rigorously validated the effectiveness of this revised approach using current clamp methods (Supplementary Fig. 11).

Supplementary Figure 11. Chemogenetic inhibition of neuronal excitability in LEC neurons.

(a) Top: Representative trace illustrating changes in membrane potentials recorded from a mCherry-expressing LEC neuron. Bottom: The mean membrane potentials were not significantly affected by application of 10 μ M of CNO ($n = 6$). (b) The same as in (a) but for hM4Di-expressing neurons. Application of CNO significantly hyperpolarized the membrane potentials in LEC neurons ($n = 6$). *** $P < 0.001$ determined by a two-tailed paired t-test. (c) Top: Representative traces showing spikes induced by current injection (160 pA) in a mCherry-expressing neuron. Bottom: Application of 10 μ M of CNO did not significantly affect number of spikes in response to current steps in mCherry-expressing neurons ($n = 7$). (d) The same as in (c) but for hM4Di-expressing neurons. Application of 10 μ M of CNO significantly reduced number of spikes in response to current steps in hM4Di-expressing neurons ($n = 6$). ** $P < 0.01$ determined by a two-way ANOVA.

Next, we examined the effects of inactivating the LEC on the spatial field stability and exploration-dependency of oePCs by computing the relative change in inferred spikes, spatial correlation, Δ COM, and Difference Index for each oePC between pairs of sessions. The CNO treatment did not induce significant changes in exploration-dependent neuronal activity, spatial correlation, Δ COM, or DI values of oePCs in mCherry mice, serving as a control (Fig. 7a-c). However, in hM4Di mice, we found that the neuronal activity during exploration bouts was significantly reduced after receiving CNO treatment but not saline. The CNO treatments also significantly impaired the spatial correlation, Δ COM and DI of oePCs compared to saline (Fig. 7d-f). In the revised manuscript, we have updated all relevant behavioral and imaging experiments, along with the results related to hM4Di modulation (supplementary Fig. 12).

Figure 7. Neural circuit mechanisms underlying exploration-selective coding in oePCs.

4) The inhibitory interneurons, though a small fraction in the CA1 pyramidal layer, may still contribute to the calcium signal analyzed here. The authors may want to discuss this issue.

Response: We acknowledge the point raised regarding the possible contribution of inhibitory interneurons in the hippocampus to the calcium signals in neurons expressed with GCaMP6f under the control of the hSyn promoter. Previous studies have indeed indicated that that certain CA1 interneurons exhibit spatial modulation (McNaughton BL, et al., 1983; Kubie JL, et al., 1990; Wilent WB and Nitz DA, 2007). Nevertheless, these interneurons typically display continuous firing patterns across the entire spatial track, with broader place fields compared to excitatory neurons (Frank LM, et al., 2001; Wilent WB and Nitz DA, 2007). This point has now been introduced in page 4 (Line 120-133).

To validate that oePCs primarily consist of excitatory neurons, we conducted an examination of the presence and properties of oePCs by using the AAV2/9-CaMKII α -GCaMP6f, which predominantly infect excitatory neurons. Our data show that 84.4% of neurons labeled with CaMKII α -GCaMP6f meet the criteria of place cells, with 10.1% of total cell population identified as oePCs. Notably, the percentages of both place cells and oePCs among CaMKII α -GCaMP6f labeled neurons are higher than those observed in hSyn-GCaMP6f labeled mice, which were 50% for place cells and 6.3% for oePCs. Importantly, the spatial characteristics, center of mass, and difference index of oePCs remain consistent between CaMKII α -GCaMP6f and hSyn-GCaMP6f labeled mice (Supplementary Fig. 4). These findings collectively support the inference that the majority of oePCs primarily consist of excitatory neurons.

Supplementary Figure 4. Identification of oePCs in excitatory principal neurons. (a) Images showing a representative experiment in CaMKII α -GCaMP6f-expressing CA1 neurons, including the maximum projection from original Ca²⁺ recording frames (top) and the distribution of iePCs (red) in SFP (bottom). (b) Lap-average activity of oePCs in exploration and non-exploration laps across all mice, sorted by place field centers. Spike rates were normalized to the peak firing rate of each neuron and sorted according to angular position. The right panel shows difference between the activity map in exploration and non-exploration laps. (c) Difference in spatial information between exploration and non-exploration laps in oePCs. Statistical significance was determined by a two-sided Wilcoxon signed rank test. (d) distribution of COM in cPCs (n = 263) and oePCs (n = 36).

(e) Difference index in cPCs and oePCs, respectively. Box plots show the median (horizontal line), 25–75% range (box) and outliers (whiskers). Statistical significance was determined by two-sided Mann-Whitney U tests; *** for P < 0.001.

5) The paper by Nieh EH et al., (Nature. 2021 Jul;595(7865):80-84.) provide strong support for this current study. The authors may want to integrate it into the discussion.

Response: We have cited and discussed this paper in page 9 (Line 367-386).

Reviewer #2 (Remarks to the Author):

The study examines how place cells respond while animals traverse a track that contains object cues and a reward at the end. It presents a large sample of place cells recorded using calcium imaging and specifically examines how these cells respond to periods of exploration around the objects. These periods are termed voluntary exploration.

The data are convincing in showing that place cells encode exploratory behaviour differently to non-exploratory behaviour in the same spatial location. This type of change to place cell activity as a result of the change to behavioral context within a consistent spatial location has been shown many times before with the classic example being the splitter cells from Wood et al. 2000. Since then, lots of studies have shown that place cells will encode different behavioral contexts (potentially intentions) in the same spatial location. For example, the Dudchenko lab has shown that different place cells will encode the intended destination of an animal at the start of the maze that has multiple different potential destinations.

Response: We highly appreciate the reviewer's insightful comments on our study. We agree that many studies have demonstrated the existence and properties of splitter cells or place cells that are modulated by different intended routs or destinations. These studies commonly involved animals facing a choice of different journeys, with differentially modulated place cells being recorded at or before the choice point. These important studies, including those by the Dudchenko group and other researchers, have revealed that place cells not only encode the animal's current location but also predict future destinations.

In the present study, the task was designed to ensure that the mice consistently followed the same paths. While the animals had the option to either explore the object or refrain from exploration, their future route and destination remain constant. Furthermore, we systematically ruled out the possibility of mice moving to a different destination when approaching the object by excluding the off-track locations. As a result, the sole distinguishing factor in task contingencies affecting place cells' activity is the exploratory intention. We hope that the Reviewer will concur that our findings suggest a new role of hippocampal neurons in encoding the exploratory intention, particularly when the influence of intended destinations is minimized.

In the current experiment the authors suggest that behaviors / cellular activity based on voluntary intentions could be dissociated from behaviors / cellular activity based on rewards. They claim that their findings support a new type of code for voluntary intention that is different to splitter type activity and independent of reward. However, animals choosing to run a particular route to a reward are demonstrating a voluntary intention. It could also be argued that exploration of an interesting object is rewarding. This presents 2 issues: firstly that it isn't clear why rewarded vs. non-rewarded intentions would be coded differently and secondly what is the nature of reward? While I think the current study presents clear and convincing evidence of place cell modulation by behavioral context, I don't think it provides evidence for reward independent voluntary behavior. Indeed, I don't think such a class of behavior would be likely.

In the example at the beginning of the paper voluntary intention is illustrated by a situation where we stop to get a newspaper. However, the newspaper in this example is a reward. In the current task the animals stop to explore an object and so a better metaphor might be stopping to look at a statue. Again though - are we sure this isn't rewarding? Even if we accept that it isn't why are reward based behaviors not driven by voluntary intention? These 2 issues need to be explained.

Response: We agree that the exploratory behaviors in rodents may not align with the concept of voluntary behaviors (see General Response, pages 1-2). We have revised the Introduction in response to these comments accordingly. Nevertheless, we maintain our proposition that encoding exploratory intentions is distinguishable from neural representations in response to food rewards. Detailed responses and evidence supporting this conclusion is provided below for all specific points raised.

Specific issues to address in the paper:

Please illustrate the 24 angular bins in 1C. This might be the quality of the image but it looks like the diagrams have more than 24 bins?

Response: We appreciate the reviewer's observation regarding Fig. 1c and the overall analysis in the paper. We acknowledge the oversight in the initial manuscript regarding the resolution and the lack of bin information. In our revised version, we have submitted all figures in high resolution, including Figure 1c, and have now included the number of bins in Figure 1c and other relevant figures. We believe these changes will enhance the clarity and precision of the presented data.

Some aspects of the analysis are a little circular. Exploratory bouts are scored manually as periods where the animal slows to explore the object. Figures 1E and 1F show that 'intentional exploration' is slower than

non-exploration and that this demonstrates clear differences between the 2 types of behaviour. However, non-exploration is defined as animals showing “no apparent slowing” on line 557 and so the data in 1 E & F represent only that the analysis has worked, not that there are 2 classes of behaviour. Similarly, intentional exploratory bouts are defined by their proximity to objects (lines 553-554) and Figures 2D&E just show that this method worked, not that intentional behaviour is grouped around objects as suggested here: “Compared to cPCs, whose place fields were evenly distributed across the entire track, the place fields of iePCs were maximally distributed with proximity to the locations of objects, as shown in the activity map sorted according to place field centers (Fig. 2D, E) and distribution of the center of mass (COM) (Fig. 2F)”. The analysis only permits intentional behaviour to be found around objects.

Response: We acknowledge the ambiguity in our previous statement and appreciate the reviewer’s diligence in pointing it out. We have revised the relevant sections in the Method session to enhance clarity and eliminate any potential issues related to circular logic.

We have revised the section as following (Line 652, Page 22): Behavioral bouts were defined as the period from when the mouse entered the $1/4\pi$ (in radians) in the track prior to the object until the $1/4\pi$ passing the object. The exploration and non-exploration behaviors were meticulously annotated by blinded, trained observers on a frame-by-frame basis. It was characterized by instances where mice engaged in sniffing, whisking, or touching the object with their nose or forepaws, or when they ambulated directly towards the object. Non-exploration was characterized by instances where mice traversed past the object without exhibiting discernible interaction with the object. The validity of the classification of the annotated behavioral bouts was subsequently confirmed by evaluating the difference in behavioral variables and the speed-time relationship from each bin (Fig 1e-f and Supplementary Fig. 1). Behavioral bouts with ambiguous classification were excluded from further analysis. Sequencing of exploration and non-exploration laps was depicted in Supplementary Fig. 13.

We have revised the section as following (Line 712, Page 23): “The oePCs were identified from test sessions, each consisting of a minimum of three exploration and three non-exploration bouts recorded for at least one of the objects. Each behavioral bout covered $1/2 \pi$ of the track, spanning $1/4 \pi$ prior to the object to $1/4 \pi$ after the object. For each object, the maximum binned activity was computed for each exploration and non-exploration bouts, and the difference in mean between the two behaviors was determined.” For each object, because $1/4$ of the entire track is analyzed, the oePC fields could possibly be distributed across any location within this $1/4$ circle. Contrary to this possibility, our results reveal a clustered distribution of the center of mass near the object, rather than a uniform distribution across the entire $1/4$ circle. We hope that the revised section in the Method will help to clarify this issue.

It would be interesting to illustrate the 2SD in rho on 3A/B.

Response: We apologize for the unclear image caused by compression. The mean + 2SD in Rho is depicted as dashed black lines in Figure 3a.

The authors propose that iePCs and reward cells are fundamentally different classes of cells. The current setup doesn't allow this distinction to be made. They show clearly that cells that respond to the reward location do not respond to the object locations. However, Figure 1E shows that iePCs typically only respond to one of the object locations. Using the same logic you could say that iePCS that respond at object position 1 are fundamentally different to all other cells recorded. To demonstrate that these cells are not reward cells the reward would need to be presented in the object locations and a lack of activity would need to be demonstrated. At present the data show clearly that populations of place cells respond to all of the interesting areas in the arena and that each interesting area is represented by different place cells. However, they do not show that rewards are represented differently to other interesting areas.

Response: We appreciate the reviewer for raising this important question. In response, we conducted a set of experiments aimed at investigating whether oePCs remain active when an object is substituted with a food reward at the same location. During a recording session, we initiated the experiment by allowing the mice to freely explore the familiar object, as described in our original experimental design. This enabled the identification of oePCs from both the object exploration and non-exploration laps. In a subsequent lap, we removed the existing object and introduced a food pallet at the exact same location before the mouse reached it (referred to as the "reward lap"). In the following laps, a reward was released only when the mouse exhibited pausing and exploration behaviors at the same location, with approximated a 50% chance determined at random (referred to as the "reward-expecting laps"). It is important to note that neuronal activity was computed during the period that mice are approaching the food reward, and the periods of the food consumption were excluded from the analysis related to reward expectation.

As indicated by the results (Fig. 4g-j), the previously identified oePCs displayed significantly reduced activity during the reward-expecting laps, notwithstanding the presence of evident anticipation behaviors characterized by significant deceleration prior to reaching the site and increased time spent in that location. Taken together, these findings indicate that oePCs display low activity to food rewards, thereby suggesting a clear differentiation between these cells and reward-associated cells as distinct cell types.

Fig. 4: (g) The experimental design diagram illustrating the within a recording session, the experiment was initiated by allowing the mice to freely explore the familiar object, enabling the identification of oePCs from both the object exploration and non-exploration laps. In a subsequent lap, we removed the existing object and introduced a food pallet at the exact same location before the mouse reached it (referred to as the “reward lap”). In the following laps, a reward was released only when the mouse exhibited pausing and exploration behaviors at the same location, with approximated a 50% chance determined at random (referred to as the “reward-expecting laps”). It is important to note that neuronal activity was computed during the period that mice are approaching the food reward, and the periods of the food consumption were excluded from the analysis related to reward expectation. (h) Significant difference in moving speed change (left) and investigatory time (right) were found in exploration or reward-expecting bout compared to non-exploration bouts (30, 77 and 37 laps, respectively; 6 experiments; 5 mice). * $P < 0.01$ determined by Kruskal-Wallis H Tests. (i) Gaussian-smoothed activity maps of a representative oePC (top) and lap-average activity of pooled oePCs (bottom) before and after the object was replaced by food rewards. (j) Mean activity of all oePCs during the exploration, non-exploration, and reward periods ($n = 23$ cells, 5 mice). The activity of oePCs during exploration periods is significantly higher than the other conditions (***, $P < 0.001$, Kruskal-Wallis H Test).

The most convincing argument that the iePCs are object oriented is the fact that their activity is disrupted by inhibiting LEC input. It would be informative to see whether reward cells are also inhibited in the LEC condition. If they are not this would be good evidence that iePCs are not reward related.

Response: We addressed this question by the following experiments. First, we tested the efficiency of hM4Di AAVs sourced from several different vendors by using whole-cell patch clamp techniques in LEC slices. Consequently, we adopted an approach of co-infusing AAV2/9-hsyn-Cre and AAV2/9-DIO-hM4Di-mCherry for chemogenetic inhibition, and the effectiveness of this revised approach was validated using patch clamp methods (Supplementary Fig. 11). Next, we performed all relevant behavioral and imaging experiments to examine whether oePCs and reward-associated cells were sensitive to CNO in the hM4Di- or mCherry-expressing animals. In the revised manuscript, our data indicate that inhibition of LEC significantly decreased the activity of oePCs during exploration (Fig. 7). In contrast, LEC inhibition did not significantly affect the activity of reward-associated cells in the same group of experiments (Supplementary Fig. 12). These data demonstrate that oePCs and reward cells are differently modulated by the inhibition of LEC.

Figure 7. (d-f) The oePCs showed similar degree of changes in spatial correlation (0.64 (0.44 – 0.83) and 0.32 (-0.01 – 0.57), $P < 0.01$) or ΔCOM (0.18 (0.07 – 0.55) and 1.36 (0.27 – 2.31), $P < 0.05$) after saline injection ($n = 24$) and CNO injection ($n = 22$ in 5 mice), respectively. DI values after injection of CNO (0.28 (0.01 – 0.58)) were significantly different from the other conditions, including before saline (0.89 (0.83 – 0.94)), after saline (0.77 (0.57 – 0.88)), or before CNO (0.85 (0.76 – 0.91)), $P < 0.001$. The statistical significance between cumulative curves was determined using a two sample Kolmogorov-Smirnov test. The statistical significance between two groups or multiple groups was determined using two-sided Mann Whitney U Test or Kruskal-Wallis H Test, respectively, and is indicated by asterisks: *** for $P < 0.001$, ** for $P < 0.01$, and * for $P < 0.05$.

Supplementary Figure 12. Effects of CNO on reward-associated cells in mCherry and hM4Di expressing mice. (c) Application of CNO does not significantly affect activity of reward-related neurons in mCherry-expressing mice. Left: Activity maps of a representative cell that responds to reward site (indicated by “R”) before and after CNO treatment. Right: Inferred activity of reward-related cells after treatments of saline ($n = 21$ cells) or CNO ($n = 14$ cells). (d) The same as in (c) but for hM4Di-expressing mice. NS, no significance as determined by two-sided Wilcoxon signed rank tests.

Reviewer #3 (Remarks to the Author):

In this study, the authors employed miniature endoscope imaging to record the activity of a population of dorsal CA1 neurons during an intentional exploration task. They found that a small population of CA1 neurons not only encode for the spatial information but also the animal’s investigatory intention. Furthermore, inhibiting the LEC maintained the spatial properties of these neurons but reduced the activity between exploration and non-exploration intention. Together, the authors suggest that this newly identified group of CA1 neurons could be critical in bridging external information (place fields) with internal intention (exploration).

The study is definitely interesting and novel, and the results will bring a new perspective to the area of hippocampal place field. Many control parameters and different maze settings were conducted to test the properties of these iePCs, which is a lot of work considering each experiment is combined with in vivo

calcium imaging. However, more detail about the behavior task and the analyses should be provided. In addition, a lot of the figures were not labelled, which makes the manuscript a bit difficult to read/understand. Below I listed the questions that I had.

Response: We thank the reviewer for these supportive comments.

Major:

1. What's the purpose of the square- and circular-shaped maze? It seems like most of the analyzes were done in the square maze or did the authors combine all the trials?

Response: We employed two distinct maze shapes for two primary reasons. First, subsequent to our initial identification of oePCs within the square-shaped maze, we aimed to verify whether these cells could be observed in diverse environmental contexts. Second, we observed that within the square maze, the mice's positions during exploratory laps may exhibit slight deviations from the track compared to non-exploratory laps, a phenomenon likely influenced by the square maze's shape. A circular-shaped maze effectively mitigates this issue. In Fig. 2, data were presented separately for square and circular maze. In the subsequent analysis, such as in Fig. 3 and 4, data were combined from both mazes.

2. It would be nice if the authors could describe the behavioral paradigm in detail.

- In the methods, it only says 'animals reached ~85% rate of correct-direction laps over all laps in about 10 days, etc....'. How many laps does the animal run on the imaging days.

Response: Each mouse underwent a daily 20-minute training session and reached an ~85% rate of correct-direction laps over all laps in approximately 10 days. The number of laps per session during training periods varied depending on the proficiency and motor ability of each animal. We have revised the relative parts in Methods to improve the clarity (Line 621-624, Page 21).

- '1-3 different objects were placed'  how many objects were placed exactly? If more than one, is the neuronal response for each object combined for the analyses?

Response: In most of our experiments, two or three objects were positioned at distinct location. However, for the sake of brevity and clarity in the illustrative figures (e.g., Fig 1e, 3a, 4a, 7), we depicted only a single object as the target object for analysis. It is essential to emphasize that in actual experiments two or three objects were indeed present. The exploratory and non-exploratory laps were independently annotated for each object. For example, during a lap, a mouse might explore the object at position 2 but not the one at position 3. In such a case, we annotated this lap as an exploratory bout for object 2 while a non-exploratory bout for object 3.

During sessions involving three objects, we observed a trend where mice displayed reduced exploratory behaviors at location 4 (as depicted in the illustration), which is the area just prior to the reward site (location 1). We attribute the reduced exploration at location 4 to its proximity to the reward site, as it often led the mice to make a direct dash toward location 1 to obtain the reward, thereby bypassing exploration at location 4. Consequently, in some of our experiments, we chose to place objects solely at location 2 and 3.

The only experiment in which the animals were subjected to a single-object test is the emergence of oePCs in Supplementary Fig. 3. This choice was made to prevent potential interference with place fields associated with adjacent objects during early periods of oePC emergence.

- ‘Imaging sessions were performed every day for 1-2 sessions with 20min ITI’  how many days were imaged? For the 2 sessions within a day, was the object placed at the same location? If the animals are familiar with the object and maze, does that decrease their exploration trials? If yes, does the activity change between session 1 and session 2 within the same day?

Response: We depict the general behavioral paradigm as the following figure. As hereinbefore described, in our standard protocol, data analysis primarily focuses on the recordings from the initial test sessions to identify exploration-dependent cells. If the animal’s behavior did not meet the set criteria (a minimum of three exploration and three non-exploration bouts recorded), additional recording sessions were scheduled for subsequent days as needed. The test sessions typically have a duration of approximately 10 to 15 minutes. After the identification of these cells, further experiments were conducted, including assessments of cross-day stability, object displacement, object replacement, object removal, transitioning to novel maze, and more. Among these experiments, those involving transitioning to novel maze and chemogenetic manipulations were conducted with two imaging sessions daily, with a 30-minute inter-session interval. These methodology details have been revised in the Methods section (Line 627-648, Page 21). The animals did not exhibit a decrease in the percentage of exploration trials, and the stability of oePCs was relatively maintained, as supported by experiments involving saline treatment in mCherry-expressing mice (Fig. 7 and Supplementary Fig. 12)

- What location was the reward given? Was it at a random location and how far away was it from the 3 objects?

Response: As shown in the above diagram, reward is delivered a fixed position (location 1).

- All of these factors could affect the authors’ interpretation on ‘exploration’ and ‘intention’.

3. While it is nice the authors showed the location of iePCs in each animal in SFig 2, the image quality that I got from the PDF file is not very good. Can the authors provide a summary table of how many oePCs are identified in each animal? Moreover, do the authors observe a trend on whether these cells have a specific location in CA1 or are they intermingled among other place cells?

Response: We observed an intermingled distribution of oePCs among CA1 cells, without a specific pattern of distribution. In the revised manuscript, we enlarged these images to improve the resolution. Furthermore, we have included the detailed information on the number of oePCs identified in each animal (see the revised Supplementary Fig. 2).

4. I'm confused about this 'off-track' position. It looks like the authors can better decode the behavior or oePC activity when they included off-track trials, so why do they keep saying they are 'excluding' them (line 125)?

Response: We apologize for any confusion originated from the original manuscript's wording. Briefly, in the off-track positions, oePCs might respond to different positions instead of the exploration behaviors. Consequently, after excluding the off-track positions, the remaining data show indistinguishable Rho positions between exploration and non-exploration laps. A detailed revision of the results section has been implemented to reflect this clarification (Line 136-169, Page 4-5).

5. Figure 5A – I presume this is done on expert animals? Did the authors follow the same cells and track their place fields when the object was moved away? Also, how many trials were performed for each new distance?

Response: The experiments were done on expert animals. The same cells were tracked within the same recording session when the object was moved away. Please see Supplementary Fig. 13d for detailed number and sequence of exploratory and non-exploratory laps.

6. Figure 6B, how many exploration trials are in early vs. late? Do they have similar numbers, or do the early sessions have a lot more exploration trials?

Response: Please see Supplementary Fig. 13b for detailed number and sequence of exploratory and non-exploratory laps in these experiments.

Day-to-day dynamism

7. Figure 6D, does replacing the max box induce new oePCs within the imaging field? It doesn't seem like it does, judging from Fig 6C, but why is that?

Response: Replacing the box resulted in the emergence of new oePCs within the imaging field. However, we initially identified the oePCs from the familiar box and subsequently tracked these cells when the animal was introduced in the new box. Approximately 58% of total cells remained active in a different maze, with 71% of oePCs remained active in both contexts. Among these, 11 out of 15 oePCs still exhibited spatial modulation, while 4 cells were not modulated by spatial or intentional information in the novel context. None of the cells met the criteria for being identified oePC in the new context. We have included these additional details on page 8 (Line 288 -298) of the revised manuscript.

We have discussed a possible explanation for the remapping in oePCs (Line 367-386, Page 9). As contextual information is stored as independent representations for a multitude of environments, a change in context likely reflects a hidden state inference of the animal's subjective judgement of the environment (Sanders H et al., 2020). Each distinct hidden state can be encoded as an individual state space (Stachenfeld KL et al., 2017), with internal variables such as abstract values or navigation intentions being encoded in different dimensions together with other physical variables related to the context. Consequently, state space change leads to extensive alterations in the neural coding of all variables across dimensions within the hippocampus.

Minor:

1. Figure 2A, B – missing lap #

Response: The lap numbers have been added.

2. Figure 2D, E – the heat bar doesn't have a unit.

Response: We have now included the units in the figure.

3. Figure 2F – there are no y-axis title for the spatial information and difference index.

Response: We have now included the Y-axis title in the figure.

4. Figure 4A should be shown like Figure 3D because it'll provide speed information as well. And it's missing the y-axis unit.

Response: We have modified Figure 4A in a similar form of Figure 3D which includes the speed information. The Y-axis has been added.

5. Is firing rate the same as spike rate (mentioned in figure 1)? And why is it 'au'?

Response: All the neuronal activity was computed using CASCADE and reported using the raw output of inferred firing probability multiplied by the frame rate. As recommended in the CASCADE GitHub repository (<https://github.com/HelmchenLabSoftware/Cascade>), CASCADE can be applied to 1P calcium data extracted with CNMF-E. However, the recommended units should be "arbitrary units" instead of "spike rate", since the extracted traces cannot be properly transformed into real dF/F values in these cases. We have now replaced "firing rate" into "inferred activity" in arbitrary units. This modification has been further elucidated in the revised manuscript (Line 664-675, Page 22).

6. Figure 6A and 6C and in general, more labels on the figures will help the readers to understand the plots easier without having to read the figure legends every single time. What are the two heat plots in the middle?

Response: We apologize for the confusion resulting from the absence of figure labels. The information of each panel has been added in the revised manuscript.

Reviewer #4 (Remarks to the Author):

This paper explores the intriguing possibility that the hippocampus contains neurons that respond to the animals' investigatory intentions. The authors use calcium imaging to record from area CA1 in the mouse during object exploratory and non-exploratory laps. The main finding is that a percentage of neurons in CA1 becomes active before exploratory behavior at specific locations and display significant reductions in activity during non-exploratory laps (the authors call these cells intentional exploration-dependent place cells, iePC). The authors show that the majority of iePC cells are not the same as those responding to reward location or object identity. For example, the cells display similar levels of activity when the objects are replaced in the same locations and, on average, the cells did not show increased activity after an object was removed (shown only in 9 cells, 3 mice, Figure S3). Finally, the authors show that inhibition of the lateral entorhinal cortex, diminishes the activity mismatch between exploratory and non-exploratory laps. Although the authors propose an interesting idea, the data do not support the claim that these cells reflect "voluntary intentions neurons". Alternative explanations need to be ruled out. Below I summarize the main points that need to be addressed to clarify the meaning and impact of the findings.

Response: We thank the Reviewer for the thorough and comprehensive review of our manuscript. The insightful comments and constructive suggestions, which encompassed methodological details, data analysis, and the overall writing of the paper, have proven to be very valuable in enhancing the quality of

our research.

1. The use of the term “voluntary intentions” for the interpretation of rodent behavior is an overreach. The definition of voluntary intentions can be extremely subjective and, as pointed out in the literature, subject to philosophical biases (Libet , 1985). Voluntary intentions have been proposed not only to require endogenous responses, but also reflect a level of introspective awareness. This manuscript does not offer compelling evidence that mice have a level of cognitive abstraction for this process. Therefore, calling hippocampal cells “object- exploratory cells” would be a more cautious and appropriate way to represent the findings.

Response: We thank the reviewer for bring up this important concern. We acknowledge that mouse behaviors are primarily driven by instinct, learned associations, and responses to environmental cues, all of which are linked to biological needs. These behaviors differ from what we typically attribute to “voluntary intentions” or a level of introspective awareness like those of humans. Therefore, exploration or investigatory behaviors in rodents represent a form of intended behavior rather than the kind of voluntary, intentional behaviors associated with conscious planning, abstract thinking, or complex cognitive processes. Accordingly, we have removed the terms “voluntary intentions” and “voluntary behaviors” from the manuscript in the revised version. And in the revised manuscript, we have changed iePC into “object-exploration-dependent place cells” (oePC) as a more cautious way.

Nevertheless, we would like to propose that while mice do not process the same level of cognitive complexity or cognitive abstraction as humans, they do exhibit a range of behaviors that involve abstract thinking or intention. For instance, theta oscillations in rat hippocampus contain information about volitional navigation intentions and motivational importance of the animal (AM Wikenheiser and D Redish, 2015). And another recent study has shown that the mouse hippocampus can encode abstract variables during navigation and incorporating these abstract variables into existing representations of physical variables (EH Nieh et al, 2021, as suggested by Reviewer 1). These findings have been discussed and incorporated into the Discussion section of the revised manuscript (Line 367-386, Page 9).

2. Along these lines, the introduction states that voluntary intension cells could code “what we want to do” (line 24). Implying voluntary intension in the mouse is deceiving, not only because it is not known if mice are capable of voluntary intention (point above), but also because some alternatives have not been ruled out (see points below).

Response: The responses have been consolidated in the subsequent paragraph.

3. The authors claim that that the oePC cells are not reward cells because these cells (at least most of them) are not the same neurons that fire at the reward locations, suggesting that their role is independent of reward properties. The authors fail to note that there are several kinds of reward and different cells may respond to distinct reward characteristics. A subpopulation of cells may respond to edible rewards, whereas another subset may respond to object exploration. Different hippocampal cells have been shown to code reward locations, action value of reward, anticipation of reward, etc. Therefore, it is possible that distinct subpopulations also represent edible vs. object exploration reward value.

Response: The responses have been consolidated in the subsequent paragraph.

4. Additionally, hippocampal cells have been shown to fire in anticipation of reward (Lee et al. 2006; Jarzeowski et al., 2022). Since the iePC cells increase activity before the mouse reaches the objects (line 140), a crucial analysis would be to compare how the same cells respond when mice are approaching the food reward (before they reach the reward location rather than at the reward location).

Response: In Fig. 4, the activity of oePCs or reward cells during reward periods were indeed computed specifically during the periods when mice were approaching the food reward, and the periods of the food consumption at the reward location were excluded. We have updated Fig. 4a to provide a clearer illustration of the relationship between the decrease in speed and the increase in oePC activity during both object exploration and food anticipation, addressing the suggestion made by Reviewer #3. We apologize for the unclear description in the original manuscript and this point has been clarified in the revised version.

We highly agree with the reviewer that in our original experimental settings we cannot exclude the possibility that oePCs and reward cells both code reward but at different locations. In response, we conducted a series of experiments to investigate whether oePCs could remain active when an object was substituted with food rewards at the same location. During a recording session, we initiated the experiment by allowing the mice to freely explore the familiar object, as described in our original experimental design. This enabled the identification of oePCs from both the object exploration and non-exploration laps. In a subsequent lap, we removed the existing object and introduced a food pallet at the exact same location before the mouse reached it (referred to as the “reward lap”). In the following laps, a reward was released only when the mouse exhibited anticipation behaviors at the same location, with approximated a 50% chance determined at random (referred to as the “reward-expecting laps”). It is important to note that neuronal activity was computed during the period that mice were approaching the food reward, and the periods of the food consumption were excluded from the analysis. As indicated by the results (Fig. 4g-h), the previously identified oePCs displayed significantly reduced activity during the reward-expecting laps, notwithstanding the presence of evident anticipation behaviors characterized by significant deceleration prior to reaching the site and increased time spent in that location. These findings indicate that oePCs display low activity to food rewards.

Figure 4 (g) The experimental design diagram illustrating the within a recording session. (h) Significant difference in moving speed change (left) and investigatory time (right) were found in exploration or reward-expecting bout compared to non-exploration bouts (30, 77 and 37 laps, respectively; 6 experiments; 5 mice). * $P < 0.01$ determined by Kruskal-Wallis H Tests. (i) Gaussian-smoothed activity maps of a representative oePC (top) and lap-average activity of pooled oePCs (bottom) before and after the object was replace by food rewards. (j) Mean activity of all oePCs during the

exploration, non-exploration, and reward periods (n = 23 cells, 5 mice). The activity of oePCs during exploration periods is significantly higher than the other conditions (***, P < 0.001, Kruskal-Wallis H Test).

Furthermore, in the revised manuscript, our results have shown that inhibition of LEC by chemogenetic approaches significantly reduced the activity of oePCs without affecting the activity of reward-associated cells (Supplementary Fig. 12). Taken together, these data indicate a clear differentiation between these cells and reward-associated cells as distinct cell types.

Supplementary Figure 12. Effect of LEC inhibition on RA cells. (c) Application of CNO does not significantly affect activity of reward-related neurons in mCherry-expressing mice. Left: Activity maps of a representative cell that responds to reward site (indicated by "R") before and after CNO treatment. Right: Inferred activity of reward-related cells after treatments of saline (n = 21 cells) or CNO (n = 14 cells). (d) The same as in (c) but for hM4Di-expressing mice. NS, no significance as determined by two-sided Wilcoxon signed rank tests.

5. One of the main points of the paper is that iePC cells reflect voluntary intention because their activity increases only when animals slow down to explore the objects. It would be important to see the sequence of exploratory and non-exploratory laps in different mice. Do non-exploratory laps occur more often after animals have explored the objects substantially? Although the authors give 3 days of habituation, it is different to see the objects early on the first day of testing vs. at the end of the testing session. Are exploratory laps observed during the object habituation period (3 days prior imaging)? How do the authors rule out that the non-exploratory laps are not the result of habituation to the objects rather than the voluntary intension to omit exploration? Rodents running in mazes containing objects or performing object place tasks show decreased exploration for the objects and places that are familiar (Manns and Eichenbaum, 2009). Therefore, a distribution of non-exploratory laps over time of testing will be important.

Response: The sequencing of exploratory and non-exploratory laps within each recording session is detailed in Supplementary Fig. 13a. These sequences reveal a random distribution of exploratory laps rather than a notable increase in non-exploratory laps in the later phase of the session.

Furthermore, we have recorded the percentage of exploratory laps relative to the total laps during four consecutive days, and our analysis revealed no significant decline in the occurrence of exploratory laps (Supplementary Fig. 3c). These collective findings support that the occurrence of non-exploratory laps cannot be attributed to habituation to the objects.

Supplementary Figure 3 (c) The percentage of exploratory laps relative to total laps does not show significant difference across Day 1 to 4. Dots with the same color indicate the same mouse (N = 5 mice). Statistical significance was determined by Kruskal-Wallis H tests.

6. It is not clear why some animals were tested with 1 object and other with 3. Did the authors do some analysis to compare animals trained with 1 vs. 3 objects? Are the iePC cells specific to one object or multiple objects in sessions with 3 objects? Are there more exploratory laps observed in animals trained with 3 objects? For example, in Figure 1A the square and cylinder schematic contain 3 objects; however, Figure 1E shows an example of an exploratory and non-exploratory cell with 1 object. How did the authors decide when to use 1 vs. 3 objects? Why the method was not consistent across animals?

Response: In most of our experiments, two or three objects were positioned at distinct location. However, for the sake of brevity and clarity in the illustrative figures (e.g., Fig 1e, 3a, 4a, 7), we depicted only a single object as the target object for analysis. We would like to clarify that in actual experiments two or three objects were indeed present. The exploratory and non-exploratory bouts were independently annotated for each object. For example, during a lap, a mouse might explore the object at position 2 but not the one at position 3. In such a case, we annotated this lap as an exploratory bout for object 2 while a non-exploratory bout for object 3. The only experiment in which the animals were subjected to a single-object test is the emergence of oePCs in Supplementary Fig. 3. This choice was made to minimize possible interference from adjacent objects during early periods of oePC emergence.

During sessions involving three objects, we observed a trend where mice displayed reduced exploratory behaviors at location 4 (as depicted in the illustration), which is the area prior to the reward site (location 1). We attribute the reduced exploration at location 4 to its proximity to the reward site, as it often led the mice to make a direct dash toward location 1 to obtain the reward, thereby bypassing exploration at location 4. Consequently, in some of our experiments, we chose to place objects solely at location 2 and 3

7. The authors rule out that object identity/novelty could trigger the differential activity between exploratory and non-explorative laps because, when objects are replaced, no differences were observed between familiar and replacement conditions. When were the new objects replaced during testing? How much time elapsed between a regular session and one with object replacements?

Response: In all object replacement experiments, the replacement of the object was performed in the middle of a recording session. A detailed depiction of list of laps before and after the object replacement has been included in Supplementary Fig. 13c.

How different were the objects (a picture will be useful)?

Response: The image of all objects utilized in our experiments has been incorporated in Supplementary Figure 7a.

I found this experiment interesting because the results do not show a change in overall activity. However, I wondered if this experiment was conducted early during testing or after the animals were extremely familiar with the objects. Without knowing the specifics, it is impossible to determine the relevance of these findings.

Response: The experiments involving the replacement of novel objects were conducted after animals had become extremely familiar with these objects, i.e., following three days of training sessions and a minimum of one day of recording sessions. We agree with the reviewer that the lack of neural differences observed in this experiment was unexpected. Initially, this experiment was conducted with a group of four mice. In an effort to further corroborate these findings, we replicated the experiment with an additional two animals. The updated results, comprising data from all six animals, continue to demonstrate no significant alterations in the spatial correlation, Δ COM or difference index in oePCs between the familiar and objects. We have integrated the results from all animals and have updated Fig. 5c and d in the revised manuscript accordingly.

Figure 7 (c) Lap-average activity of oePCs during exploration and non-exploration laps and activity difference during the mice's visits to the same box with the familiar object replaced with a novel object at the same location. Neuronal activity was normalized to the peak activity of each neuron and sorted by place field centers according to angular position across mice. (d) Spatial property changes in oePCs during object replacement compared to the control condition: oePC spatial correlation, 0.84 (0.76 – 0.91), $P = 0.59$; ΔCOM , 0.15 (0.05 – 0.32), $P = 0.82$. The difference index was 0.88 (0.79 – 0.94) before and 0.89 (0.81 – 0.92) after object replacement ($P = 0.55$, $n = 36$ in 6 mice). The statistical significance between cumulative curves was determined using a two sample Kolmogorov-Smirnov test. The statistical significance between two groups or multiple groups was determined using two-sided Wilcoxon rank sum test or Kruskal-Wallis H Test, respectively, and is indicated by asterisks: *** for $P < 0.001$, ** for $P < 0.01$, and * for $P < 0.05$.

Moreover, we took into consideration that most objects used in the initial experiments were geometric solids, as referred to as “simple objects” in the image above. This raised the possibility that the animals might not adequately distinguish between these objects. To address this, we conducted a new set of experiments adhering to the same experimental procedures but using more complex objects. The results remained consistent with those obtained in previous experiments tested with simple objects, exhibiting consistent spatial correlation and ΔCOM in oePCs relative to the control. Notably, the only discernible variation was a minor yet statistically significant decrease in the DI values, with medians of 0.8626 (familiar objects) versus 0.8033 (novel complex objects), as determined by a two-sided Wilcoxon signed rank test ($P = 0.023$, Supplementary Fig. 7b-c); but the DI values after object replacement were still substantially higher than 0 ($P = 2.85 \times 10^{-6}$).

Supplementary Figure 7. Effect of replacing familiar objects with novel complex objects. (a) Representative

objects used in the experiments. **(b)** Lap-average activity of oePCs during exploration and non-exploration laps, and activity difference upon replacing the familiar object with a novel complex object at the same location. Inferred activity was normalized to the peak activity of each neuron and sorted by place field centers according to angular position across mice. **(c)** Changes in spatial properties of oePCs during object replacement: spatial correlation, 0.86 (0.63 – 0.90); Δ COM, 0.13 (0.07 – 0.72). The difference index was 0.86 (0.75 – 0.92) before and 0.80 (0.72 – 0.84) after object replacement ($P < 0.05$ determined by a two-sided Wilcoxon signed rank test, $n = 29$ in 4 mice).

It is quite established that animals will explore new objects more than familiar ones, therefore, the lack of neural differences in this experiment is intriguing and surprising.

Response: We also acknowledge that object replacement is expected to alter the encoding of object information in hippocampal neurons. Accordingly, we extended our investigation to classic PCs (cPC) with place fields near the object in the same experiments. Interestingly, we observed significant changes in the fields of cPCs in response to the replacement of the novel object. Illustrated in Supplementary Fig. 7d are one oePC and one cPC as representative cells from an experiment of simple object replacement (see below). The oePC exhibited selective activation during exploration, with its field remaining consistent after the object change. In contrast, a cPC displayed similar fields during both exploration and non-exploration, but exhibited significantly altered fields after object replacement. Supplementary Fig. 7e depicts an oePC and a cPC from another experiment involving complex object replacement, revealing similar results. Taken together, these findings suggest that object replacement induces more profound changes in the activity of cPCs compared to oePCs, highlighting the relative insensitivity of oePCs to object identity or novelty. Our results support the notion that oePCs play a role in representing exploratory intentions, which may operate independently of the specific information associated with the objects themselves.

Supplementary Figure 7. (d) Substantial alterations were observed in the activity maps of a representative cPC (right), whereas the activity of a representative oePC (left) from the same experiment remained consistent during object replacement. (e) Similar to (d), but for the replacement of a complex object.

8. The object elimination described in Figure S3 included only 9 cells in 3 mice. Considering that there is some variability in the results, why did the authors only select a subset of mice/cells?

Response: In the previous experiments, we carried out the experiments of object elimination only in three animals. We would like to clarify that we did not deliberately select a subset of mice or cells for representation. However, we have since increased the number of mice by carrying out additional experiments, resulting in a larger sample size of oePCs (oePCs=22, mice=6; Supplementary Fig. 8). Our data demonstrate that among 22 cells, only 2 cells exhibited increased activity similar to “misplace cells”. Thus, our results continue to affirm that oePCs do not exhibit activation following the removal of the object.

Supplementary Figure 8. The activity of iePCs before and after object removal. (b) Pooled data show that, after the removal of the object, firing rates in 20 out of 22 iePCs (black) were similar to those observed during non-exploration bouts, whereas 2 out of 22 iePCs exhibited increased activity (red) similar to the property of “misplace cells”. Inferred activity was averaged for exploration or non-exploration bouts, and shown for the first three laps after the removal of objects. All neural activity was normalized to the mean activity during exploration in each cell ($n = 6$ mice). Statistical significance was determined by Kruskal-Wallis H tests. *** for $P < 0.001$ compared to the inferred activity during exploration laps.

9. It is well known that mice find some objects more interesting than others. In sessions with 3 objects, does object preference interact with iePC cell activity? Since preferred objects should generate more exploratory behaviors than non-preferred ones, it would be interesting to determine if the DI index is affected for preferred vs. unpreferred objects.

Response: In order to investigate whether DI index is affected by object preference, we calculated the percentage of laps during which a given object was explored in a recording session. We then computed the correlation coefficient between this preferred score and DI of each oePCs across all experiments. Our analysis revealed no significant correlation between these two variables, indicating that DI index is not affected by the animal's preference for a particular object. This result has been incorporated in Supplementary Fig. 1f for reference.

Supplementary Figure 1. (f) No significant correlation was observed between the percentage of exploratory laps relative to the total laps, reflecting animals' object preference, and the difference indices.

10. It is really confusing to know precisely what the timeline/design of the experiment was. The Caption of Figure 1A mentions 9 experiments. Does this include sessions in the square and the cylinder across days? How long was each session and how many days were recorded? The method states that after the period of object habituation (3 days), imaging sessions were performed every day for 1-2 sessions, but it is not clear how many days the testing involved.

Response: The general timeline of the experiments is illustrated in the following diagram (Supplementary Fig. 1a). The test sessions typically have a duration of approximately 15 minutes. In our standard protocol, data analysis primarily focuses on the recordings from the initial test sessions to identify exploration-dependent cells. If the animal's behavior did not meet the set criteria (a minimum of three exploration and three non-exploration bouts recorded), additional recording sessions were scheduled for subsequent days as needed. After the identification of these cells, further experiments were conducted in parallel groups, including assessments of cross-day stability, object displacement, object replacement, object removal,

transitioning to novel maze, and more. Notably, to prevent potential alterations in the properties of already identified oePCs due to object manipulation, each of the aforementioned experiments was conducted with separate groups of animals in most cases. For example, data from two out of the nine experiments depicted in Fig. 1 were included in the cross-day experiments. Consequently, the specific duration of testing was determined based on the requirement of each experiment.

11. The results should specify that the cells were tracked across sessions. The authors only clarify that neurons were tracked across sessions in the method. The fact that this explanation is not provided early on, leaves many points unclear. Are the 107 iePC cells the ones that have SFPs across sessions? Did the authors analyze iePC cells that were temporary active (cells that show exploratory activity only during one session but could not be tracked across days)? If so, are the temporary and consistently active iePC cells similar? How many cells were tracked relative to the total number of cells recorded? The free tracking algorithms sometimes capture artifacts, were some adjustments made to the code to make sure that the tracked cells were neurons?

Response: Among the 15 experiments that were used to identify the 107 oePCs, two experiments were also included in the cross-day analysis. In the cross-day recording experiments as reported in Fig. 6, we only reported oePCs that were tracked in both sessions (referred to as consistently active oePCs). In these experiments, we further analyzed “temporarily active” oePCs and observed no significant difference in their difference index compared to the “consistently active” oePCs. These cells did not display significant responses to reward locations either. Please see Supplementary Fig. 10 (shown below) for the results.

In these experiments, on average 60.0% of total cells from an imaging session can be tracked to those in another session, and 67.7% of total oePCs can be tracked from the same experiments. All tracked ROIs from pairs SPF images were plotted and manually inspected for quality.

The same neurons that were activated across two sessions were identified by the SFP correlation using the cell registration method as previously reported. The method used SFP from the early-sessions as a reference map, and aligned with this map the SFP from the post-sessions after correction of position offset and rotation. Then the method models the distribution of centroid distances for neighboring cells from early and post sessions and gets their weighted sum to determine whether they are the same cells or different cells. The method provides a P_{same} registration threshold that is optimized to the dataset of each mouse. In our experiments, a pair of cells was considered to have the same identity if $P_{\text{same}} \geq 0.5$. The centroid distance between a pair of cells which had the same identity was less than 12 μm . All tracked ROIs from pairs SPF images were plotted and manually inspected for quality. This information has been included in Line 676-685, Page 22.

Supplementary Figure 10. Comparisons between consistently and temporarily active oePCs. (a) Aligned SPF maps overlaid from two recording sessions, one day apart. Yellow cells in the right panel indicate consistently active oePCs that can be tracked across both days, while red cells represent temporarily active oePCs observed on the first day. In all paired cross-day recordings, 60.0% of total cells remained active in recording sessions on subsequent days, while 67.7% of oePCs remained active. (b) Lap-average activity of oePCs in exploration and non-exploration laps across all mice, sorted by place field centers. Spike rates were normalized to the peak firing rate of each neuron and sorted according to angular position. The right panel shows difference between the activity map in exploration and non-exploration laps. (c) For temporarily active oePCs, spatial information during exploration laps (median 0.94 (25th–75th percentiles 0.66 – 1.14)) was significantly higher than during non-exploration laps (0.36 (0.25 – 0.63)) ($n = 17$ cells, 8 mice). However, spatial information during each behavior type was indistinguishable between consistently and temporarily active oePCs. Statistical significance was determined by a Kruskal-Wallis H test. (d) No significant difference was observed in the difference index between consistently (0.5 (0.81 – 0.91), $n = 34$) and temporarily active oePCs (0.89 (0.82 – 0.95), $n = 17$), as determined by a two-sided Mann-Whitney U test. (e) Both consistently and temporarily active oePCs show significantly lower activity during reward periods compared to during object exploration periods, as determined by Kruskal-Wallis H tests.

12. I found the remapping observed after object displacement counterintuitive to the authors' argument. The results show that object displacement caused significant remapping and changes in the place fields' center of mass, with most fields disappearing when the object was moved 4 cm. If the iePC cells were indeed voluntary intention cells, then they should not have been affected by object displacement. Using the example provided by the authors, if someone wants to stop to buy the New York Times along his/her path to work, it should not matter if the location of an ambulatory seller changes from one day to another. The intention to stop and buy the newspaper should be the same regardless of the location of the seller. However, if the cells are merely coding object/place locations, then remapping makes a lot of sense.

Response: Our findings indicate that oePCs represent spatial information in an exploration-dependent manner. First, these cells are distinct from the neurons that encode only intentions, like previously reported cells that are more active when the animal approached objects across different locations (Wood ER et al., 1999). Instead, oePCs found in our studies exhibit specific place fields, i.e. they are active only during exploration at specific locations. Therefore, oePCs are different from pure "intention cells".

Second, we aimed to test whether these cells solely encode object/place locations or the presence of an object at a location. We designed a set of supplementary experiments as following: we positioned the

object outside of the track and obstruct its visibility using opaque partitions. After being trained in this behavioral arena and having well learned the location of the object, the mice were allowed to freely explore the concealed object by passing through a designated door or to bypass the door location without engaging in exploration (Supplementary Fig. 6). It is worth emphasizing that the mice were unable to visually perceive the object unless they chose to pass through the door and engage in exploration.

Next, we excluded the data corresponding to the “off-track” points, which denote the points where the mice passed through the door and went outside of the track. Employing the same criteria for identification of oePCs in the manuscript, we proceeded to identify oePCs and uncover a group of neurons that specifically increase activity before exploration of the objects. Their activity maps clearly show that these cells were activated prior to the mice reaching the designated door and before they directly visualize the presence of the object (Supplementary Fig. 6b). These oePCs exhibit properties consistent with those identified in the initial experiments (Fig. 2f). These data collectively provide evidence that oePCs encode the intention of object exploration rather than visual perception of the object’s presence or property. Therefore, we hope the reviewer will align with our view that these cells indeed reflect exploratory intentions rather than merely representing physical information about the object/location itself.

Supplementary Figure 6. Encoding of exploratory intentions by oePCs for indirectly visible objects.

(a) The experimental design diagram illustrating the object is positioned outside of the track with visibility obstructed by opaque partitions. Following training in this behavioral arena with objects, mice freely explored the concealed object by passing through a designated door or to bypass the door location without engaging in exploration. (b) Top view of the maze displaying the location of a sample object and rewards (R). The dashed curves represent the mouse’s trajectory during an exploration lap, with the red segment indicating “off-track” positions. (c) Color-coded neuronal activity of an example oePCs along the mouse’s trajectory, with each panel illustrating a lap during exploration (top) or non-exploration (bottom). These activity maps demonstrate increased oePC activity exclusively during exploration bouts, even when the object’s location was well-learned but not directly visible. (d) Lap-average activity of oePCs in exploration and non-exploration laps across all mice, sorted by place field centers. Spike rates were normalized to the peak activity of each neuron and sorted according to angular position. The right panel shows difference between the activity map in exploration and non-exploration laps. (e) The spatial index during exploratory laps (0.72 (0.50 – 0.88)) were significantly higher than non-exploratory laps (0.56 (0.26 – 0.7)) in identified oePCs (n = 22). (f) The difference index of oePC (0.89 (0.81 – 0.95)) was significantly higher than that of classic PCs (0.02 (-0.25 0.27), n = 310).

13. The differential activity of iePC cells during exploratory vs. non-exploratory laps disappears when animals are placed in a new context (Figure 6D). Are the iePC cells normal place cells when animals are moved to a new box? If the subpopulation of iePC cells does not show the same properties across contexts, it would be important to discuss this characteristic in the discussion and provide a possible explanation.

Response: In the revised manuscript, we have included the following information (page 8, Line 288 -298): Approximately 58% of total cells remained active in a different maze, with 71% of oePCs remained active in both contexts. Among these, 11 out 15 oePCs still exhibited spatial modulation, while 4 cells were not modulated by spatial or intentional information in the novel context.

We have discussed a possible explanation for the remapping in oePCs. As contextual information is stored as independent representations for a multitude of environments, a change in context likely reflects a hidden state inference of the animal's subjective judgement of the environment (Sanders H, et al., 2020). Each distinct hidden state can be encoded as an individual state space (Stachenfeld KC, et al., 2017), with internal variables such as abstract values or navigation intentions being encoded in different dimensions together with other physical variables related to the context. Consequently, state space change leads to extensive alterations in the neural coding of all variables across dimensions within the hippocampus (added in Line 367-386, Page 9).

14. The protocol to train and test the animals is confusing. According to the method, mice were first trained to run on tracks in either a square or a cylinder in one direction before starting recordings. Then, animals received 1 or 2 sessions (line 539) with the miniendoscope attached. Why some animals received 1 and others 2 sessions? How long were the sessions? How did these 2 sessions generate 9 experiments mentioned in line 344? How many days the cell activity was recorded? Why were different groups of mice tested in the cylinder and square? Having a schematic of the training timeline and design in one of the Figures will be useful.

Response: We apologize for any confusion stemming from the original manuscript's wording. As hereinbefore described, the general timeline of the experiments is illustrated in the diagram provided (Supplementary Fig. 1). In our standard protocol, data analysis primarily focuses on the recordings from the initial test sessions to identify oePCs. If the animal's behavior did not meet the set criteria (a minimum of three exploration and three non-exploration bouts recorded), additional recording sessions were scheduled for subsequent days as needed. The test sessions typically have a duration of approximately 10 to 15 minutes. After the identification of these cells, further experiments were conducted, including assessments of cross-day stability, object displacement, object replacement, object removal, transitioning to novel maze, and more. Notably, to prevent potential alterations in the properties of already identified oePCs due to object manipulation, each of the aforementioned experiments was conducted with separate groups of animals in most cases. The nine experiments mentioned in line 344 (Fig. 1) were recorded from nine mice, with one recording session allocated to each mouse.

We employed two distinct maze shapes for two primary reasons. First, subsequent to our initial identification of oePCs within the square-shaped maze, we aimed to verify whether these cells could be observed in diverse environmental contexts. Second, we observed that within the square maze, the mice's positions during exploratory laps may exhibit slight deviations from the track compared to non-exploratory

laps, a phenomenon likely influenced by the square maze's shape. A circular-shaped maze effectively mitigates this issue.

15. The authors refer to firing rate when describing calcium events (lines 41, 38, 588, 607, 612, 802, Figure S3, Figure S4 etc.). This is incorrect. Calcium imaging only provides an indirect measure of activity. I commend the authors for using CASCADE, but they should refer to “inferred spikes” or “inferred activity” not firing rate. Did the authors do anything to normalize fluorescence across cells and days? For example, do the authors use the raw output from CASCADE or do they normalize before or after?

Response: All the neuronal activity was computed using CASCADE and reported using the raw output of inferred firing probability multiplied by the frame rate without additional normalization. In response to the reviewer’s suggestion and recommendation by the CASCADE GitHub repository (<https://github.com/HelmchenLabSoftware/Cascade>), we have now replaced “firing rate” with “inferred activity” in arbitrary units. This modification has been further elucidated in the revised manuscript (Line 664-675, Page 22).

Given that the inferred neuronal activity was not subjected to normalization, we opted to compute spatial correlation, ΔCOM and difference index to assess the changes in oePC properties. In certain experiments, such as those related to object displacement, replacement or removal (Fig. 4g, Fig. 5, Supplementary Fig. 7 and 8), we manipulated objects within the same session without interrupting the recording, ensuring that the same neurons were recorded during the entire imaging session. For experiments requiring the tracking of neurons across different sessions, such as cross-day assessments and chemogenetic manipulations, we maintained the consistency of the imaging focal plane and the power of LEC excitation. We believe that these procedures will effectively mitigate any possible influence arising from slight variations of imaging conditions.

16. To determine the significance of iePC cells the authors performed bootstrap analysis “the identity of exploration/non-exploration bouts were randomly shuffled, and the difference in mean was computed as chance data. This shuffling was repeated 10,000 times to establish the likelihood that the observed difference in mean could have emerged by chance”. 10,000 seems excessive. If 3 exploring and 3 non-exploring trials were included, there were 20 unique combinations.

Response: We appreciate the reviewer for bringing up this issue. We have calculated the number of combinations in all experiments listed in the following figure, revealing a median of 1144 (56-11628) combinations across experiments. Therefore, we acknowledge that using 10,000 iterations was indeed excessive for most experiments. In the revised manuscript, we have adjusted the number of iterations to 1,000 times. This modification in the method does not impact the previous results concerning oePC identification.

17. For any given animal and session, how many laps were exploratory vs. non-exploratory? The authors report that sessions were included if at least there were 3 of each but the total number per animal would be useful.

Response: A figure depicting the lap list is provided as shown above (also Supplementary Figure 13).

18. The difference index was computed using the difference in max bin activity for exploration and non-exploration trials divided by their sum. The maximum activity was calculated over 5 bins with the object in the center. Considering that the arena was divided in 24 binds, each bin was 15 degrees ($360/24=15$). Therefore, the max activity was computed over 75 degrees. This seems quite excessive because with 3 objects the maximum activity will cover 225 degrees of the arena. What is the actual angular distance around each object? How much does the DI change if a more restricted radial measure is used in the analysis?

Response: In accordance with the reviewer's suggestion, we have now calculated the angular distance for each object, which falls within a range of 0.11-0.33 radians (equivalent to 6.4-18.5 degrees). We agree with the reviewer that employing 5 bins for DI calculation may be excessive. Therefore, in the revised manuscript, all DI values has been recalculated using a 3-bin coverage approach.

Minor comments

-I recommend thorough editing of the manuscript. Tenses are mixed in many paragraphs, including the abstract (second sentence mixes present and past tense)

Response: We have conducted a comprehensive review of the manuscript and rectified all instances of tense inconsistency.

-line 30 should be "as they travel through that location."

Response: We have revised line 30 to read, "as they travel through that location."

-The Figure caption of Figure 3C says top and bottom and it should be left and right.

Response: We acknowledge the oversight in the Figure 3C caption and have made the necessary correction.

-The authors use probability and firing rate without defining these terms. Throughout the paper, firing rate should be replaced by inferred activity or inferred rate. Probability should be defined.

Response: We have substituted occurrences of "firing rate" with "inferred activity" where applicable.

-the design indicates that animals are familiarized to the objects prior the imaging. When is the baseline activity recorded (line 154)? Neural activity when animals decrease speed in areas outside the object regions is compared with baseline, but it was not clear when the baseline was computed.

Response: The baseline activity was calculated using the activity from all time points within laps, excluding those during exploration, reward, and other periods. Detailed methods have been included in the revised manuscript (Line 740-747, Page 24).

-in line 190 the term “control experiments” is misleading. These are control laps where even and uneven exploratory laps are correlated. It is not a separate experiment.

Response: We have revised the manuscript to replace the term "control experiments" with “the control” (Line 246, Page 7).

References:

Frank LM, Brown EN, Wilson MA (2001), A comparison of the firing properties of putative excitatory and inhibitory neurons from CA1 and the entorhinal cortex. *Journal of Neurophysiology* 86(4):2029-40.

Kubie JL, Muller RU, Bostock E (1990), Spatial firing properties of hippocampal theta cells. *The Journal of Neuroscience* 10(4):1110-23.

McNaughton BL, Barnes CA, O'Keefe J (1983), The contributions of position, direction, and velocity to single unit activity in the hippocampus of freely-moving rats. *Experimental Brain Research* 52:41-49.

Nieh EH, Schottdorf M, Freeman NW, et al. (2021) Geometry of abstract learned knowledge in the hippocampus. *Nature* 595(7865):80-84.

Sanders H, Wilson MA, Gershman SJ (2020), Hippocampal remapping as hidden state inference. *eLife* 9:9:e51140

Stachenfeld KL, Botvinick MM, Gershman SJ (2017), The hippocampus as a predictive map. *Nature Neuroscience* 20:1643-1653.

Wikenheiser AM, Redish AD (2015), Hippocampal theta sequences reflect current goals. *Nature Neuroscience* 18(2):289-94.

Wilent WB, Nitz DA (2007), Discrete Place Fields of Hippocampal Formation Interneurons. *Journal of Neurophysiology* 97:4152-4161.

Wood ER, Dudchenko PA, Eichenbaum H (1999), The global record of memory in hippocampal neuronal activity. *Nature* 397:613-616.

REVIEWERS' COMMENTS

Reviewer #1 (Remarks to the Author):

The authors nicely addressed my questions. The manuscript has been greatly improved and is ready for publication.

Reviewer #2 (Remarks to the Author):

The authors have addressed all of my comments.

Reviewer #3 (Remarks to the Author):

I am satisfied with the revised manuscript. The authors have adequately addressed previous concerns and provided detail explanations to the experiments and methodologies.

Reviewer #4 (Remarks to the Author):

Reviewer 4 comments.

Review of resubmission.

I am very pleased with the meticulous revision of the manuscript titled "Conjunctive Encoding of Exploratory Intentions and Spatial Information in the Hippocampus." The authors have diligently addressed all of my comments. I have only a minor additional suggestion.

In addition to a comprehensive review of all my points, I highly appreciate the inclusion of the experiment involving hidden objects, as it demonstrates that visual inputs do not elicit activity in the oePCs. However, it is acknowledged that these results do not definitively exclude the possibility that the memory of the object could have triggered the activity. In the subsequent section, the authors present additional analyses illustrating that the activity of oePCs does not lead to potentiation, in line with the findings of Monaco et al. (2004). I propose that the authors integrate these findings towards the conclusion of the object occlusion experiment. If memory were the catalyst for oePC activity, one would expect potentiation, given its association with episodic memory (Monaco et al., 2004). The absence of potentiation in their observations reinforces the authors' assertion that oePCs play a crucial role in intentional exploration.

Minor points

I still found some typos

Line 121 should read: also have been “shown”

Line 270-271 move “Supplementary Fig. 9” to the end part of the sentence to facilitate reading.

Line 289: replace “remained” with remaining.

Line 344 replace “was” with “being”: “with the only different variable being the animal’s intention”

REVIEWERS' COMMENTS

Reviewer #1 (Remarks to the Author):

The authors nicely addressed my questions. The manuscript has been greatly improved and is ready for publication.

Response: We thank the reviewer again for the comments and suggestions, which greatly improved the quality of this manuscript.

Reviewer #2 (Remarks to the Author):

The authors have addressed all of my comments.

Response: We thank the reviewer again for the comments and suggestions, which greatly improved the quality of this manuscript.

Reviewer #3 (Remarks to the Author):

I am satisfied with the revised manuscript. The authors have adequately addressed previous concerns and provided detail explanations to the experiments and methodologies.

Response: We thank the reviewer again for the comments and suggestions, which greatly improved the quality of this manuscript.

Reviewer #4 (Remarks to the Author):

Reviewer 4 comments.

Review of resubmission.

I am very pleased with the meticulous revision of the manuscript titled "Conjunctive Encoding of Exploratory Intentions and Spatial Information in the Hippocampus." The authors have diligently addressed all of my comments. I have only a minor additional suggestion.

Response: We thank the reviewer again for the comments and suggestions, which greatly improved the quality of this manuscript.

In addition to a comprehensive review of all my points, I highly appreciate the inclusion of the experiment involving hidden objects, as it demonstrates that visual inputs do not elicit activity in the oePCs. However, it is acknowledged that these results do not definitively exclude the possibility that the memory of the object could have triggered the activity. In the subsequent section, the authors present additional analyses illustrating that the activity of oePCs does not lead to potentiation, in line with the findings of Monaco et al. (2004). I propose that the authors integrate these findings towards the conclusion of the object occlusion experiment. If memory were the catalyst for oePC activity, one would expect potentiation, given its association with episodic memory (Monaco et al., 2004). The absence of potentiation in their

observations reinforces the authors' assertion that oePCs play a crucial role in intentional exploration.

Response: We have integrated the results of the hidden object task and the analysis from the previous reports by Monaco et al (Supplementary figures 6 and 7). Based on these revisions, we have concluded that the oePC activity was not evoked by object memory. Please see Line 179-204, page 6-7 for the revised content.

Minor points

I still found some typos

Line 121 should read: also have been "shown"

Line 270-271 move "Supplementary Fig. 9" to the end part of the sentence to facilitate reading.

Line 289: replace "remained" with remaining.

Line 344 replace "was" with "being": "with the only different variable being the animal's intention"

Response: We are grateful for the reviewer's thorough review and have corrected all identified typos.